# Tropospheric ozone production and chemical regime analysis during the COVID-19 lockdown over Europe

Clara M. Nussbaumer[1], Andrea Pozzer[1], Ivan Tadic[1], Lenard Röder[1], Florian Obersteiner[2], Hartwig Harder[1], Jos Lelieveld[1,3], and Horst Fischer[1]

[1]Max Planck Institute for Chemistry, Department of Atmospheric Chemistry, 55128 Mainz, Germany
[2]Karlsruhe Institute of Technology, 76021 Karlsruhe, Germany
[3]Climate and Atmosphere Research Center, The Cyprus Institute, Nicosia, Cyprus

**Correspondence:** Clara Nussbaumer (clara.nussbaumer@mpic.de)

**Abstract.**

The COVID-19 (Coronavirus disease 2019) European lockdowns have led to a significant reduction in the emissions of primary pollutants such as NO (nitric oxide) and $NO_2$ (nitrogen dioxide). As most photochemical processes are related to nitrogen oxide ($NO_x \equiv NO + NO_2$) chemistry, this event has presented an exceptional opportunity to investigate its effects on
air quality and secondary pollutants, such as tropospheric ozone ($O_3$). In this study, we present the effects of the COVID-19 lockdown on atmospheric trace gas concentrations, net ozone production rates (NOPR) and the dominant chemical regime throughout the troposphere based on three different research aircraft campaigns across Europe. These are the UTOPIHAN campaigns in 2003 and 2004, the HOOVER campaigns in 2006 and 2007 and the BLUESKY campaign in 2020, the latter performed during the COVID-19 lockdown. We present in situ observations and simulation results from the ECHAM5/MESSy
Atmospheric Chemistry model which allows for scenario calculations with business as usual emissions during the BLUESKY campaign, referred to as "no-lockdown scenario". We show that the COVID-19 lockdown reduced NO and $NO_2$ mixing ratios in the upper troposphere by around 55 % compared to the no-lockdown scenario due to reduced air traffic. $O_3$ production and loss terms reflected this reduction with a deceleration in $O_3$ cycling due to reduced mixing ratios of $NO_x$ while NOPRs were largely unaffected. We also study the role of methyl peroxyradicals forming HCHO ($\alpha_{CH_3O_2}$) to show that the COVID-19
lockdown shifted the chemistry in the upper troposphere/tropopause region to a $NO_x$ limited regime during BLUESKY. In comparison, we find a VOC limited regime to be dominant during UTOPIHAN.

## 1 Introduction

COVID-19 (Coronavirus disease 2019) describes the disease accompanying an infection with the SARS-CoV-2 (severe acute respiratory syndrome coronavirus-2) virus. The disease is highly infectious and can have severe health consequences, including
premature death, particularly for elderly and people with pre-existing conditions (WHO, 2021). On 11 March 2020, COVID-19 was declared a pandemic by the World Health Organization (WHO, 2020a,b). As a response, in many countries worldwide - including the European continent - governments initiated a shutdown of the daily life for minimizing the spread of the virus, which is referred to as COVID-19 lockdown. Among others, this included a reduction in vehicular and industrial activities as

well as sharp restrictions on air travel accompanied by a reduction in atmospheric pollutants such as nitrogen oxides ($NO_x \equiv$ $NO + NO_2$) (Venter et al., 2020; Kroll et al., 2020; Chossière et al., 2021; Onyeaka et al., 2021; Salma et al., 2020; Matthias et al., 2021; Forster et al., 2020).

NO and $NO_2$ are important atmospheric trace gases as they are involved in almost all photochemical processes taking place in the earth's atmosphere. $NO_x$ directly impacts the production of tropospheric ozone ($O_3$) which is a hazard to human and plant health (Nuvolone et al., 2018; Mills et al., 2018). Together with volatile organic compound (VOC) oxidation, NO forms $NO_2$ within the $HO_x$ cycle, catalyzed by an OH radical. Under the influence of sunlight, $NO_2$ can subsequently form $O_3$ through the reaction with molecular oxygen as shown in Reaction (R1) (Leighton, 1961; Crutzen, 1988; Lelieveld and Dentener, 2000; Pusede and Cohen, 2012; Pusede et al., 2015; Nussbaumer and Cohen, 2020).

$$NO_2 + O_2 \xrightarrow{h\nu} NO + O_3 \qquad\qquad\qquad\qquad \text{(R1)}$$

Various termination reactions such as the formation of $HNO_3$ from OH and $NO_2$ or other radical recombinations cause ozone chemistry to be non-linear, which means that a reduction in ambient $NO_x$ can either increase or decrease $O_3$ production (Calvert and Stockwell, 1983; Pusede et al., 2015). For low ambient $NO_x$ levels, a $NO_x$ reduction usually causes a decrease in $O_3$ production which is referred to as a $NO_x$ limited chemical regime. In contrast, a $NO_x$ reduction increases $O_3$ production when a VOC limited chemical regime is dominant - usually at high ambient $NO_x$ levels (Sillman et al., 1990; National Research Council, 1991; Pusede and Cohen, 2012). In the transition region between the two regimes, changes in $NO_x$ do not (or only slightly) impact $O_3$ production rates (Wang et al., 2018). Earlier studies on evaluating correlations of $NO_x$ and $O_3$ in the troposphere include Liu et al. (1987); Logan (1985) and Lin et al. (1988), reporting a non-linear dependence that varies with ambient levels of hydrocarbons and $NO_x$.

Many different methods enable the determination of the dominant chemical regime, such as the use of the weekend ozone effect which considers the response of $O_3$ to $NO_x$ reductions on weekends or the ratio of HCHO to $NO_2$ with various approaches from in situ observations, remote sensing and model simulations (e.g. Jin et al. (2020); Pusede and Cohen (2012); Nussbaumer and Cohen (2020); Duncan et al. (2010)). We have recently shown that the fraction $\alpha$ of methyl peroxyradicals ($CH_3O_2$) forming formaldehyde (HCHO) in correlation with ambient NO concentrations is capable of indicating the dominant chemical regime based on three different field campaigns across Europe in Finland (HUMPPA 2012), Germany (HOPE 2012) and Cyprus (CYPHEX 2014) (Nussbaumer et al., 2021). $CH_3O_2$ formed from e.g. the oxidation of acetaldehyde ($CH_3CHO$) or methane ($CH_4$) can either react with NO or OH radicals to form HCHO or undergo the competing reaction with $HO_2$ to form methyl hydroperoxide ($CH_3OOH$). For more details, please see Figure 1 in Nussbaumer et al. (2021). $\alpha_{CH_3O_2}$ consequently depends on the ambient concentrations of NO, OH and $HO_2$ and the respective rate constants for the reaction with $CH_3O_2$, the latter of which were taken from the IUPAC Task Group on Atmospheric Chemical Kinetic Data Evaluation (2021). Self-

reaction of $CH_3O_2$ as a contributor to $CH_3O_2$ loss forming HCHO is negligible. The calculation of $\alpha_{CH_3O_2}$ is presented in Equation (1).

$$\alpha_{CH_3O_2} = \frac{k_{CH_3O_2+NO} \times [NO] + k_{CH_3O_2+OH} \times [OH]}{k_{CH_3O_2+NO} \times [NO] + k_{CH_3O_2+OH} \times [OH] + k_{CH_3O_2+HO_2} \times [HO_2]} \tag{1}$$

Low values for $\alpha_{CH_3O_2}$ with a high response to NO are an indicator for a $NO_x$ limited regime whereas high values for $\alpha_{CH_3O_2}$ with little response to changing NO represent a VOC limitation (Figure 11 in Nussbaumer et al. (2021)). Investigating the dominant chemical regime is an important method for analyzing photochemical processes and air quality.

Previous studies have explored changes in air quality, trace gas emissions and the dominant chemical regime during the COVID-19 lockdown in Europe. Menut et al. (2020) reported $NO_2$ reductions between 30 and 50 % for various Western European countries in the course of March 2020 with both decreasing and increasing $O_3$ concentrations in response, depending on the location, based on surface in situ observations and model simulations. Ordóñez et al. (2020) observed decreased $NO_2$ and increased $O_3$ concentrations in Central Europe in March and April 2020 based on in situ observations compared to 2015 - 2019. While they found $NO_2$ reductions to be mainly attributed to the COVID-19 lockdown, $O_3$ enhancements were predominantly affected by meteorological changes. Chossière et al. (2021) presented evidence on $NO_2$ reductions during the COVID-19 lockdown in Europe and $O_3$ changes dependent on the dominant chemical regime through investigation of $HCHO/NO_2$ ratios based on in situ and satellite observations. Similar studies were performed by Matthias et al. (2021); Mertens et al. (2021); Balamurugan et al. (2021); Grange et al. (2021) and many more. Besides the changes within the dominant chemical regime through $NO_x$ reductions, i.e. increasing ozone within a VOC limited regime and decreasing ozone within a $NO_x$ limited regime, the COVID-19 lockdown could have potentially changed the dominant chemical regime from VOC to $NO_x$ limited as pointed out by Kroll et al. (2020) and Gaubert et al. (2021). Cazorla et al. (2021) found a lockdown induced change from a VOC to a $NO_x$ limited regime in Quito (Ecuador) based on the share of precursor loss to $HNO_3$ and $H_2O_2$. The latter is dominant for $NO_x$ limited chemistry (Kleinman et al., 2001). A change from a VOC to a $NO_x$ limited regime was also reported by Zhu et al. (2021) in China based on HCHO to $NO_2$ ratios ($NO_x$ limitation for ratios above 2 according to Duncan et al. (2010)). Most of the literature on pollutant reductions during the COVID-19 lockdown focuses on near-surface air quality and only few studies consider the free troposphere. Steinbrecht et al. (2021), Chang et al. (2022) and Bouarar et al. (2021) reported decreases in $O_3$ concentrations in the free troposphere based on in situ observations and modeling studies in the northern hemisphere. Bouarar et al. (2021) found that reduced air traffic - a unique incidence after strongly increasing aircraft activities over the past decades as shown by Lee et al. (2021) - can explain around a third of the observed $O_3$ decrease in 2020, the remaining contributions coming from ground-level reductions and meteorological differences. Reduced $O_3$ in the free troposphere was also reported by Clark et al. (2021) around Frankfurt airport. Cristofanelli et al. (2021) reported lower $O_3$ concentrations above the PBL in 2020 compared to the 1996 - 2019 average at Mount Cimone in Italy which is in line with findings by the World Meteorological Organization (2021), extended to include two mountain sites in Germany.

In this study, we present atmospheric trace gas concentrations, net ozone production rates and an analysis on the dominant chemical regime based on in situ observations during the research aircraft campaign BLUESKY which took place in May

and June 2020 over Europe, and model simulations. During this time period, aircraft activity was still strongly limited due to the COVID-19 lockdown. We compare the results to model simulations assuming business as usual emissions not impacted by government restrictions which we refer to as "no-lockdown scenario". Additionally, we present results on two previous aircraft campaigns which are UTOPIHAN (Upper Tropospheric Ozone: Processes Involving $HO_x$ and $NO_x$) in 2003/2004 and HOOVER ($HO_x$ over Europe) in 2006/2007. While many studies have been published on emissions reductions and the effect on secondary pollutants during the COVID-19 lockdown, only a few studies have investigated changes in the dominant chemical regime and to our knowledge we are the first to report a shift to $NO_x$ limited chemistry in the upper troposphere. This can demonstrate the consequences of emission changes of VOCs (including methane) and $NO_x$ on tropospheric ozone.

## 2   Observations and methods

### 2.1   Calculations of net ozone production rates (NOPR)

Besides the chemical regime, production and loss processes of $O_3$ are effective tools in exploring relevant photochemistry. As already demonstrated in Reaction (R1), $O_3$ is formed via $NO_2$ photolysis. Under the assumption of photostationary state, this term can be equated with the reactions of NO with $O_3$, $HO_2$ and $RO_2$ (Hosaynali Beygi et al., 2011). The resulting term for $O_3$ production P($O_3$) is shown in Equation (2) (Tadic et al., 2020; Leighton, 1961). $j(NO_2)$ is the photolysis frequency of $NO_2$ and $k$ describes the respective rate constant (for this work taken from the IUPAC Task Group on Atmospheric Chemical Kinetic Data Evaluation (2021)).

$$P(O_3) = [NO_2] \times j(NO_2) = [NO] \times (k_{O_3+NO} \times [O_3] + k_{NO+HO_2} \times [HO_2] + \sum_z k_{NO+R_zO_2} \times [R_zO_2]) \tag{2}$$

We assume $R_zO_2$ (the sum of all peroxy radicals) to be represented by $CH_3O_2$ which we find to be a reasonable approximation when comparing modeled $CH_3O_2$ to the overall modeled $RO_2$ as shown in Figure S1 of the Supplement, exemplarily for the BLUESKY campaign. Above 800 hPa, $CH_3O_2$ represents more than 90 % of $RO_2$. Below 800 hPa, it still accounts for more than 70 % on average. $CH_3O_2$ can be calculated via Equation (3) as derived by Bozem et al. (2017a). While the model can simulate $CH_3O_2$ mixing ratios, Equation (3) is required when working with experimental data as $CH_3O_2$ was not directly measured.

$$[CH_3O_2] = \frac{k_{CH_4+OH} \times [CH_4]}{k_{CO+OH} \times [CO]} \times [HO_2] \tag{3}$$

$O_3$ loss occurs via the reaction with NO, OH and $HO_2$ and via photolysis and can be calculated as presented in Equation (4). The photolysis of $O_3$ first yields $O^1D$ which reacts back to $O_3$ through collision with $O_2$ or $N_2$, and causes an $O_3$ loss through reaction with $H_2O$. The share of $O_3$ that is effectively lost through $O_3$ photolysis is described by $\alpha_{O^1D}$ in Equation (5) (Bozem

et al., 2017a). Additional loss due to reactions of $O_3$ with alkenes and the loss of $NO_2$ due to formation of $HNO_3$ or peroxy nitrates are negligibly small, particularly in the upper troposphere.

$$L(O_3) = [O_3] \times (k_{O_3+NO} \times [NO] + k_{O_3+HO_2} \times [HO_2] + k_{O_3+OH} \times [OH] + \alpha_{O^1D} \times j(O^1D)) \tag{4}$$

$$\alpha_{O^1D} = \frac{k_{O^1D+H_2O} \times [H_2O]}{k_{O^1D+N_2} \times [N_2] + k_{O^1D+O_2} \times [O_2] + k_{O^1D+H_2O} \times [H_2O]} \tag{5}$$

Net ozone production rates (NOPR) are then calculated from the difference in $P(O_3)$ and $L(O_3)$ whereas $P(O_3)$ can either be expressed via $NO_2$ or $NO$ reaction terms. The term $k_{O_3+NO} \times [O_3] \times [NO]$ can be neglected for the latter as it is equally present in $P(O_3)$ and $L(O_3)$.

$$
\begin{aligned}
NOPR &= P(O_3) - L(O_3) \\
&= [NO_2] \times j(NO_2) - [O_3] \times (k_{O_3+NO} \times [NO] + k_{O_3+HO_2} \times [HO_2] + k_{O_3+OH} \times [OH] + \alpha_{O^1D} \times j(O^1D)) \\
&= [NO] \times (k_{NO+HO_2} \times [HO_2] + k_{NO+CH_3O_2} \times [CH_3O_2]) \\
&\quad - [O_3] \times (k_{O_3+HO_2} \times [HO_2] + k_{O_3+OH} \times [OH] + \alpha_{O^1D} \times j(O^1D))
\end{aligned}
\tag{6}
$$

## 2.2 Field experiments

We have investigated in situ trace gas observations from three different research aircraft campaigns which are the UTOPIHAN campaigns in 2003/2004, the HOOVER campaigns in 2006/2007 and the BLUESKY campaign in 2020. Figure 1 shows an overview of the flight tracks over Europe. We have filtered the data for the tropospheric region by help of the modeled tropopause pressure (see Section 2.3) and south of 60 °N as there were no data points for the BLUESKY campaign further north. Dashed lines show the complete flight tracks during each campaign and solid lines show the data which we have considered in this study. The experimental data were obtained with a time resolution of 1 minute and subsequently adjusted to fit the model resolution of 6 minutes. For this, each sixth experimental data point (which fit the model time scale) and the data points from ± 2 minutes were averaged. The remaining data points were discarded.

### 2.2.1 UTOPIHAN 2003/04

The UTOPIHAN (Upper Tropospheric Ozone: Processes Involving $HO_x$ and $NO_x$) campaigns took place in June/July 2003 and March 2004 starting from Oberpfaffenhofen airport in Germany (48.08 °N, 11.28 °E) with the GFD (Gesellschaft für Flugzieldarstellung, Hohn, Germany) research aircraft Learjet 35A (Colomb et al., 2006; Klippel et al., 2011; Stickler et al., 2006). NO and $O_3$ were measured via chemiluminescent detection (CLD 790 SR, ECO Physics, Dürnten, Switzerland). NO data have a precision of 6.5 %, an accuracy of $\leq 25$ % and a detection limit of $< 0.01$ ppbv. $O_3$ data have a precision of 1 % and an accuracy of 5 %. $j(NO_2)$ was determined via filter radiometers (Meterologie Consult GmbH, Königstein, Germany) with a

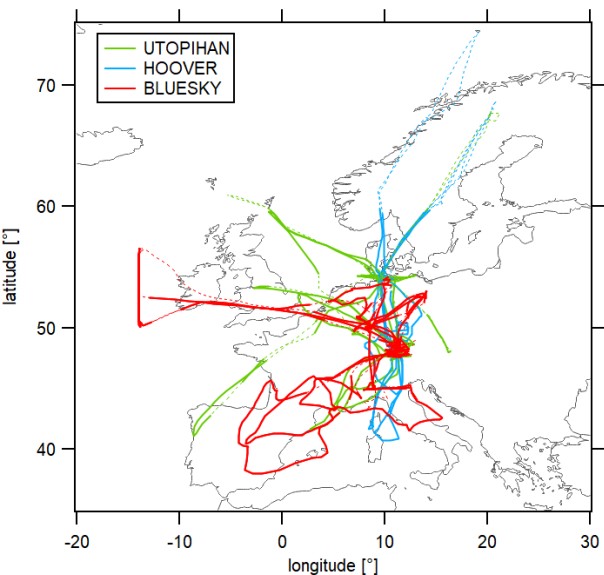

**Figure 1.** Overview of the flight tracks of the considered aircraft campaigns UTOPIHAN (2003 & 2004) in green, HOOVER (2006 & 2007) in blue and BLUESKY (2020) in red. Solid lines present the data considered in this study (filtered for troposphere and south of 60 °N) and dashed lines show the complete flight tracks.

precision of 1 % and an accuracy of 15 %. CO measurements were obtained from a tunable diode laser absorption spectrometer with a detection limit of 0.26 ppbv (30 s time resolution) and an accuracy of 3.6 % (6 s time resolution) (Kormann et al., 2005).

### 2.2.2 HOOVER 2006/07

The HOOVER ($HO_x$ over Europe) campaigns took place in October 2006 and July 2007 using the GFD research aircraft Learjet 35A with the campaign base in Hohn, Germany (54.31 °N, 9.53 °E) (Klippel et al., 2011; Bozem et al., 2017b,a; Regelin et al., 2013). NO and $O_3$ measurements were performed via chemiluminescence (CLD 790 SR, ECO Physics, Dürnten, Switzerland) with a precision of 7 and 4 %, an accuracy of 12 and 2 % and a detection limit of 0.2 and 2 ppbv, respectively (30 s time resolution) (Hosaynali Beygi et al., 2011). CO and $CH_4$ were measured via quantum cascade laser absorption spectroscopy with an accuracy of 1.1 and 0.6 % and detection limits of 0.2 and 6 ppvb, respectively (2 s time resolution) (Schiller et al., 2008). OH and $HO_2$ measurements were performed via laser-induced fluorescence with the HORUS (HydrOxyl Radical measurement Unit based on fluorescence Spectroscopy) instrument with an accuracy of 18 % and detection limits of 0.016 and 0.33 pptv, respectively (1 min time resolution) (Regelin et al., 2013). Photolysis frequencies were measured using filter radiometers (Meterologie Consult GmbH, Königstein, Germany) with a precision of 1 % and an accuracy of 15 % (1 s time resolution). $H_2O$ was measured via IR-absorption with a typical accuracy of 1 % (modified LI-6262, LI-COR Inc., Lincoln, USA) (Gurk et al., 2008; LI-COR, Inc., 1996).

### 2.2.3 BLUESKY 2020

The BLUESKY campaign took place in May and June 2020 over Europe. Eight flights were carried out using the HALO (High
Altitude Long Range) research aircraft starting from the campaign base in Oberpfaffenhofen, Germany. The goal of the campaign was to examine the effects of the COVID-19 lockdown on the troposphere and lower stratosphere over European cities, rural areas and the transatlantic flight corridor. More details can be found in Reifenberg et al. (2021) and Voigt et al. (2021). While most restrictions across Europe were in place in March and April 2020, May and June emissions, particularly from air travel but also from ground-based sources such as onroad traffic, were still affected by the COVID-19 lockdowns (Schlosser et al., 2020; Brockmann Lab, 2022; Hasegawa, 2022; EUROCONTROL, 2022). NO was measured via chemiluminescence (CLD 790 SR, ECO Physics, Dürnten, Switzerland) with a total uncertainty of 15 % and a detection limit of 5 pptv (1 min time resolution) (Tadic et al., 2020).

O$_3$ measurements were performed with the FAIRO (Fast AIRborne Ozone) instrument, which allows fast detection via chemiluminescence that is calibrated in situ by UV photometry (2.5 % combined uncertainty, 5 Hz time resolution) (Zahn et al., 2012). CO was measured via the quantum cascade laser spectrometer TRISTAR (Tracer In Situ Tdlas for Atmospheric Research) with an uncertainty of 3 % (1 min time resolution) (Schiller et al., 2008).

### 2.3 Modeling study

The modeled data were obtained from the ECHAM5 ($5^{th}$ generation European Centre Hamburg general circulation model, version 5.3.02)/MESSy2 ($2^{nd}$ generation Modular Earth Submodel System, version 2.54.0) Atmospheric Chemistry (EMAC) model which is described in Jöckel et al. (2016) and Reifenberg et al. (2021).

We use data of NO, NO$_2$, O$_3$, OH, HO$_2$, CO, CH$_4$, CH$_3$O$_2$, H$_2$O, j(NO$_2$), j(O$^1$D) temperature and pressure, modeled along the flight tracks of the described research aircraft campaigns UTOPIHAN, HOOVER and BLUESKY. The data were filtered for the troposphere using the modeled tropopause pressure. Stratospheric data were discarded. In order to evaluate the impact of reduced emissions during the COVID-19 lockdown, the model was used to simulate a scenario with usual emissions for the BLUESKY campaign which we refer to as "no-lockdown scenario". For details of the model set-up please see the paper by Reifenberg et al. (2021).

## 3 Results and Discussion

This analysis is structured as follows: As a full set of in situ observations necessary for a regime analysis and calculating net ozone production rates, which includes NO, O$_3$, OH, HO$_2$, CO, CH$_4$, H$_2$O, j(NO$_2$) and j(O$^1$D), is only available for the HOOVER campaign, we first show that the model and experimental data are in close agreement for this campaign. We conclude from this finding that the model is generally capable of reproducing the experimental data and therefore use the model data in our following analysis. In the second step, we provide a comparison between the three campaigns as well as the no-lockdown

scenario regarding the individual trace gases and net ozone production rates. We finally present our results of the analysis of the dominant chemical regime, based on $\alpha_{CH_3O_2}$.

## 3.1 Comparison of Model and Experiment

Figure 2 shows a comparison of in situ observations (orange) and model simulations (blue) for the HOOVER campaign as vertical profiles. The shaded areas present the $1\sigma$ standard deviations and the numbers of data points available for each altitude bin are shown in Table S1 and S2 of the Supplement.

Figure 2a presents the vertical profile of NO which shows the typical tropospheric C-shape distribution with the highest values at the surface (e.g. vehicle and industrial emissions) and the upper troposphere (e.g. aircraft and lightning emissions). Ground-level mixing ratios (0 - 1000 m) were around 0.4 ppbv and decreased with altitude to values of $37 \pm 27\,(1\sigma)$ pptv and $47 \pm 32$ pptv for the model and the experiment, respectively, between 3 and 9 km altitude. The only relevant deviation of model and experiment was between 10 and 11 km altitude with mixing ratios of $0.20 \pm 0.03$ ppbv and $0.39 \pm 0.32$ ppbv, respectively.

Figure 2b shows the measured and modeled $O_3$ mixing ratios which were lowest at ground levels with $43.7 \pm 14.5$ ppbv and $36.4 \pm 12.8$ ppbv for model and experiment and increased with altitude up to $128.1 \pm 22.7$ ppbv and $97.5 \pm 15.6$ ppbv, respectively. Model values were approximately 20 % higher compared to the measured data, but showed the same vertical shape. The observed positive $O_3$ bias of the modeled data is an issue almost all global models suffer from in the northern hemisphere and which has not been entirely understood yet (Revell et al., 2018; Young et al., 2018; Jöckel et al., 2016; Parrish et al., 2014).

CO vertical profiles are shown in Figure 2c which were highest at the surface with $146.4 \pm 63.2$ ppbv and $128.0 \pm 42.3$ ppbv for model and experiment, respectively, and decreased with altitude to around 70 ppbv in the upper troposphere. $HO_x$ ($\equiv$ OH + $HO_2$) are presented in Figure 2d and e. $HO_2$ mixing ratios showed a maximum value of around 20 pptv between 2 and 3 km altitude and decreased aloft to values of around 2 pptv in the upper troposphere. Model and experiment showed close agreement. OH mixing ratios were mostly below 1 pptv. Similar to NO, the main deviation between model and experiment was between 10 and 11 km altitude where measured values were higher by around 0.5 pptv. Nevertheless, the error bars representing the $1\sigma$ standard deviation of the averages overlapped at all altitudes.

Figure 2f shows the vertical profiles of $CH_4$ which did not show any particular gradient with altitude. Mixing ratios were $1809 \pm 19$ ppbv for the model simulation and $1815 \pm 40$ ppbv for the experiment throughout the campaign. $CH_4$ is needed for calculating $CH_3O_2$ via Equation (3), which we show in Figure 2g in orange compared to the model simulation of $CH_3O_2$. Figure 2h and i present the photolysis frequencies j($NO_2$) and j($O^1D$) which show close agreement for model and experiment. We show the vertical profiles for $H_2O$, temperature and pressure in Figure S2 of the Supplement. Again, model simulation can represent the experimental data well.

For the UTOPIHAN and the BLUESKY campaigns only a limited number of observations is available. Similar to the HOOVER campaigns, NO, $O_3$ and CO can be well approximated by the model simulations which we present in Figure S3 and S4 of the Supplement. Tropospheric ozone is slightly overestimated, which we attribute to the simplified representation of multiphase chemistry (clouds) in the present model version, which underpredicts chemical ozone loss (Rosanka et al., 2021).

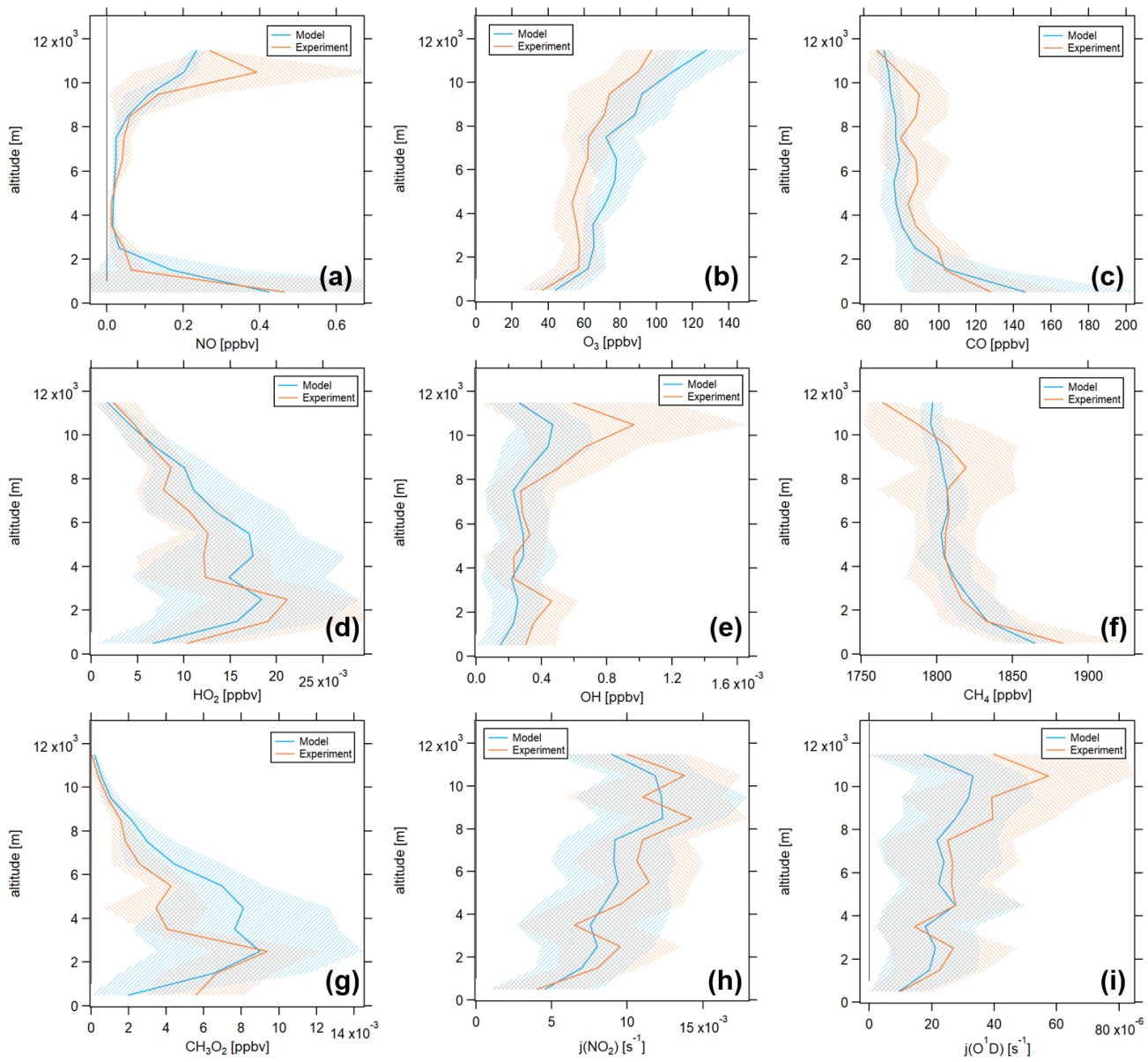

**Figure 2.** Vertical profiles of in situ observations and model data of the atmospheric trace gases (a) NO, (b) $O_3$, (c) CO, (d) $HO_2$, (e) OH, (f) $CH_4$ and (g) $CH_3O_2$ and the photolysis rates (h) $j(NO_2)$ and (i) $j(O^1D)$ during the HOOVER campaign for estimating the model performance. Blue colors show modeled data by EMAC along the HOOVER campaign flight track (Model) and orange colors show experimental data (Experiment). The orange trace in panel (g) shows the calculation of $CH_3O_2$ from experimental $CH_4$, CO and $HO_2$ via Equation (3). The shaded areas represent the $1\sigma$ standard deviation from averaging the data points at each altitude bin. The numbers of data points averaged per altitude bin are displayed in Table S1 and S2 of the Supplement.

Based on these results, we conclude that the model is generally capable of well representing the in situ observations and use the model data for all following analyses.

## 3.2 Campaign Comparison

 ### 3.2.1 Trace gases

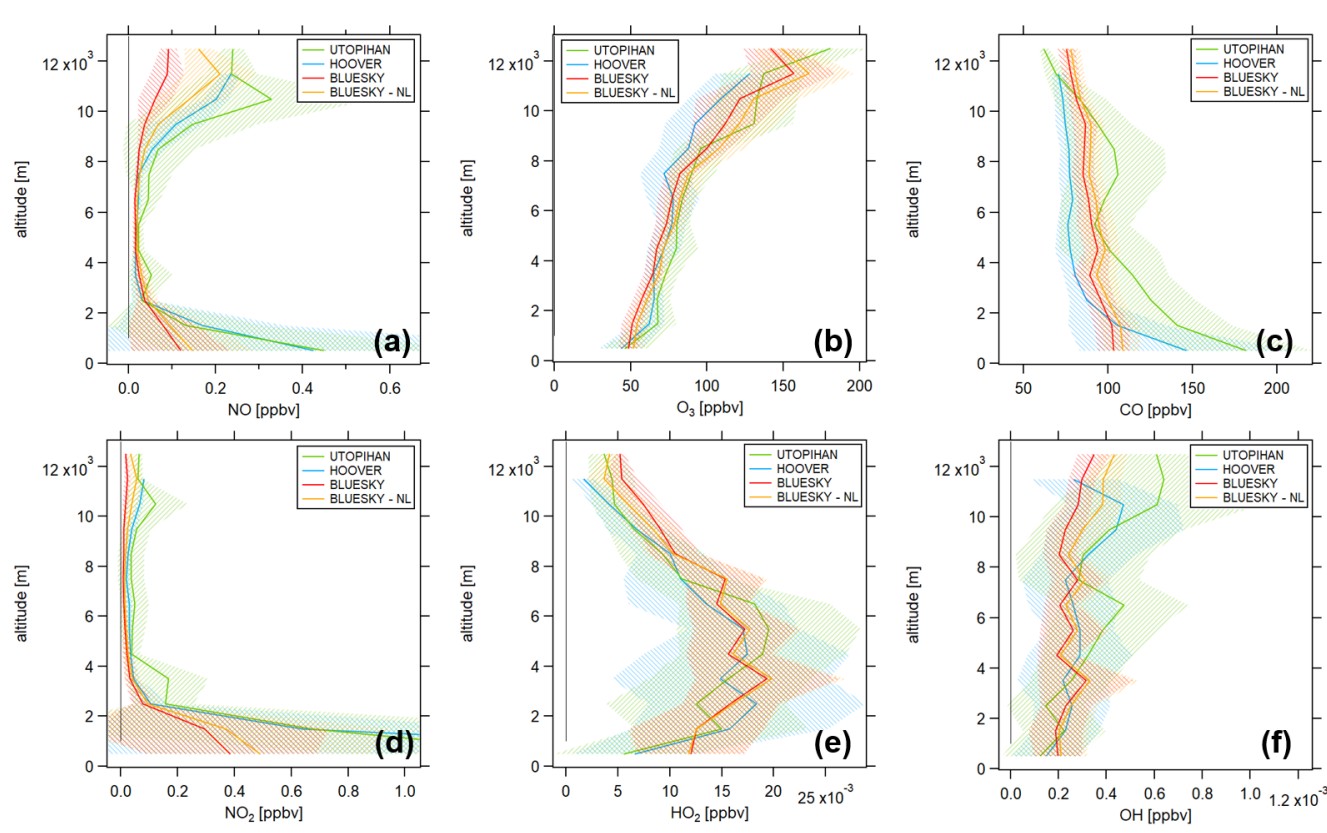

**Figure 3.** Vertical profiles of the atmospheric trace gases (a) NO, (b), $O_3$, (c) CO, (d) $NO_2$, (e) $HO_2$ and (f) OH for the campaigns UTOPIHAN (green), HOOVER (blue), BLUESKY (red) and the no-lockdown (NL) scenario (yellow). All data shown here are from EMAC model simulations along the flight track of each research campaign. Two separate simulations were run on the flight path of BLUESKY, one with lockdown and one with business-as-usual emissions. The shaded areas represent the $1\sigma$ standard deviation from averaging the data points at each altitude bin. The numbers of data points averaged per altitude bin are displayed in Table S2 of the Supplement.

Figure 3 presents the vertical profiles of some selected trace gases during the research aircraft campaigns UTOPIHAN (green), HOOVER (blue) and BLUESKY (red) which were obtained from model simulations. Yellow lines show the no-lockdown (NL) scenario for the BLUESKY campaign in 2020.

The vertical profiles of NO are presented in Figure 3a. For all campaigns, we observe the typical C-shape as described

for the HOOVER campaigns in Section 3.1. Surface (0 - 1000 m) mixing ratios were similar for UTOPIHAN and HOOVER with $0.45 \pm 0.37 \, (1 \, \sigma)$ ppbv and $0.43 \pm 0.74$ ppbv, respectively. In comparison, the ground-level concentration of NO during BLUESKY was $0.12 \pm 0.11$ ppbv. The differences in NO mixing ratios between the campaigns are the outcome of the general emission reduction due to legislative limitation of nitrogen oxides and other hazardous pollutants over the past decades, as the campaigns took place 15 - 20 years apart. We show the decrease in $NO_x$ emissions in the model over the past two decades

in Figure S5 of the Supplement. Assuming the no-lockdown scenario during BLUESKY, NO ground level mixing ratios were $0.15 \pm 0.14$ ppbv and therefore 25 % higher compared to actual mixing ratios (20 % emission reduction). This difference between lockdown and no-lockdown mixing ratios is slightly lower compared to the findings by other studies, for example by Donzelli et al. (2021) who found a NO decrease by 35 - 65 % in Valencia, Spain or by Higham et al. (2021) who reported a NO decrease by 55 % in the UK compared to 2019. A possible reason can be that the BLUESKY aircraft campaign took place

in May and June 2020 whereas the main lockdown period across Europe occurred rather in March and April. Emission were still reduced in the following months, but likely to a smaller extent. NO was low and similar for all campaigns between 3 and 8 km altitude, a region without any particular NO sources, with most values below 50 pptv. Above 10 km, NO mixing ratios were $0.29 \pm 0.19$ ppbv for UTOPIHAN, $0.21 \pm 0.03$ ppbv for HOOVER and $0.08 \pm 0.04$ ppbv for BLUESKY. In comparison, NO mixing ratios for the no-lockdown scenario were $0.17 \pm 0.08$ ppbv above 10 km altitude. This corresponds to an emission

reduction of 55 % and results in both absolute and relative NO reductions in the upper troposphere being much higher compared to ground-level reductions. The observed NO reduction in the upper troposphere can be attributed to reduced air traffic which we show in Figure 4. In addition to the vertical profiles of NO for BLUESKY (red) and BLUESKY-NL (yellow), we present the modeled BLUESKY-NL scenario without aircraft emissions in blue. In the lower troposphere, where aircraft emissions do not play a significant role, this profile is identical to the BLUESKY-NL scenario. In the upper troposphere, it is very similar to the

BLUESKY scenario (including the air travel restrictions), showing that reduced air traffic causes the observed NO decrease.

Figure 3b presents the $O_3$ vertical profiles. For all campaigns, $O_3$ mixing ratios were lowest at ground levels with values of around 50 ppbv and increased with increasing altitude up to around 140 ppbv above 10 km altitude. No significant differences between the campaigns can be observed. While ozone concentrations are dependent on various effects such as precursor levels (including $NO_x$ and VOCs) or meteorology, seasonal variations with a maximum around summer time and a minimum during

winter months are also of importance (Logan, 1985). The here shown campaigns include different seasons: the HOOVER campaigns took place in October and July and the UTOPIHAN campaigns include data from July and March. Figure S6 of the Supplement shows the vertical profiles of ozone separated into different seasons, for both modeled and measured data. Comparing late spring / early summer data of the three field campaigns reveals that $O_3$ levels during BLUESKY were lower compared to HOOVER and UTOPIHAN which is in line with findings from Clark et al. (2021), Chang et al. (2022),Bouarar

et al. (2021) and Miyazaki et al. (2021).

CO vertical profiles can be seen in 3c. Ground level mixing ratios were $181.4 \pm 39.4$ ppbv for UTOPIHAN, $146.4 \pm 63.2$ ppbv for HOOVER and slightly lower with $103.2 \pm 9.2$ ppbv for BLUESKY. Mixing ratios slightly decreased with altitude. Above 3 km altitude, CO for HOOVER was lower compared to the other campaigns (mostly between 70 and 80 ppbv). Mixing ratios

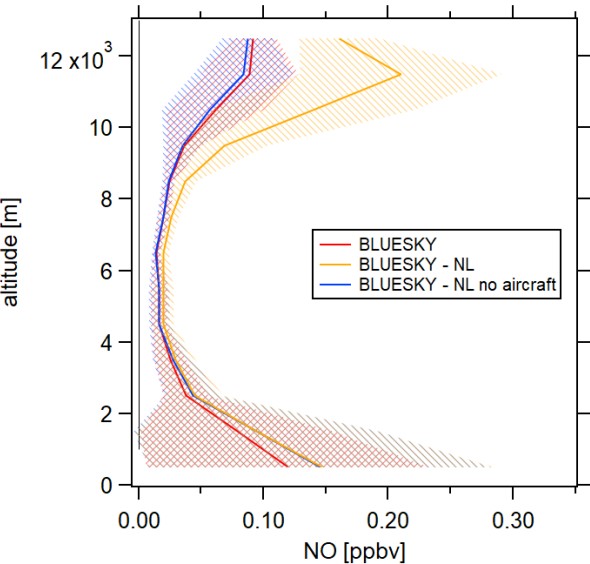

**Figure 4.** NO vertical profiles for BLUESKY (red), for the BLUESKY no-lockdown scenario (yellow) and for the BLUESKY no-lockdown scenario without aircraft emissions (blue) (model data). Upper tropospheric NO reductions observed for BLUESKY can be attributed to reduced air traffic during the COVID-19 lockdowns.

for UTOPIHAN were slightly higher up to 11 km altitude (between 90 and 110 ppbv) compared to BLUESKY (between 80
and 100 ppbv), but generally, significant differences are not evident.

Figure 3d shows the vertical profiles of $NO_2$ mixing ratios. Similar to NO, ground level $NO_2$ mixing ratios were highest for UTOPIHAN and HOOVER with $1.57 \pm 0.77$ ppbv and $2.58 \pm 2.72$ ppbv, respectively. In contrast, mixing ratios for BLUESKY were $0.39 \pm 0.30$ ppbv and $0.49 \pm 0.38$ ppbv considering the no-lockdown scenario which yields a 20 % $NO_2$ lockdown reduction, as observed for NO. We show the $NO_2$ range 0 - 1 ppbv for enabling the campaign distinction at low mixing ratios and
present the full range in Figure S7 of the Supplement. As expected for $NO_2$, mixing ratios decreased with increasing altitude. No differences between the campaigns can be observed for mid-range altitudes. In the upper troposphere, $NO_2$ mixing ratios for the individual campaigns showed the same behavior as for NO. Above 10 km altitude, $NO_2$ was on average $100.6 \pm 93.2$ pptv for UTOPIHAN, $70.5 \pm 13.5$ pptv for HOOVER and $43.1 \pm 23.1$ pptv for the no-lockdown scenario for BLUESKY. In comparison, BLUESKY $NO_2$ mixing ratios were $19.9 \pm 9.8$ pptv which corresponds to a 55 % reduction. In contrast to NO, $NO_2$
reductions were relatively higher in the upper troposphere, but absolutely higher at the surface.

Figure 3e and f show the vertical profiles of $HO_x$. $HO_2$ mixing ratios were highest at mid-range altitudes (2 - 6 km) with values up to 20 pptv and decreased aloft. OH mixing ratios were lowest at the surface (0.1 - 0.2 pptv) and increased with altitude. Above 10 km altitude, OH mixing ratios were $0.62 \pm 0.38$ pptv for UTOPIHAN, $0.40 \pm 0.24$ pptv for HOOVER, $0.30 \pm 0.06$ pptv for BLUESKY und $0.39 \pm 0.08$ pptv for the no-lockdown scenario.

 **3.2.2   Net ozone production rates**

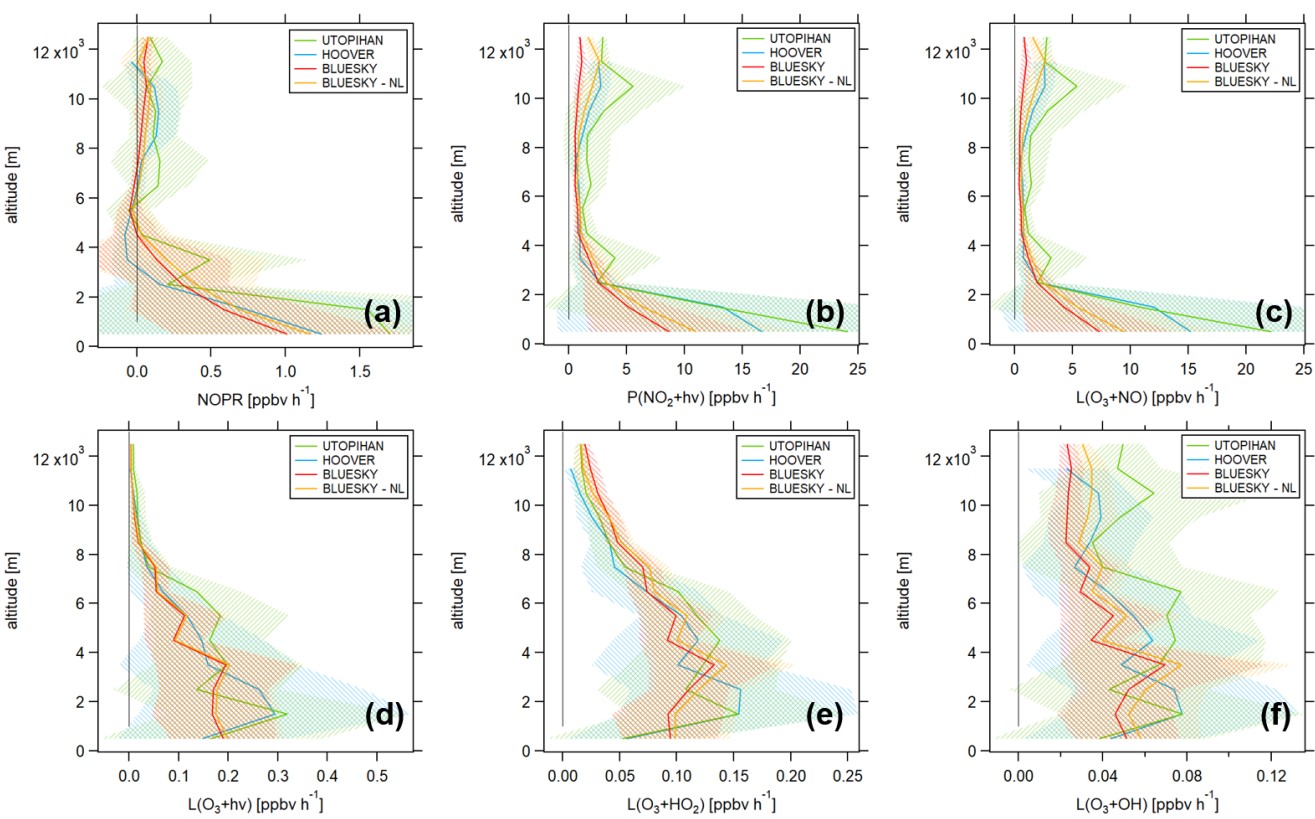

**Figure 5.** Vertical profiles of (a) net ozone production rates, (b) $O_3$ production via $NO_2$ photolysis, (c) $O_3$ loss via reaction with NO, (d) $O_3$ loss via photolysis, (e) $O_3$ loss via reaction with $HO_2$ and (f) $O_3$ loss via reaction with OH for the campaigns UTOPIHAN (green), HOOVER (blue), BLUESKY (red) and the no-lockdown (NL) scenario (yellow). The numbers of data points averaged per altitude bin are displayed in Table S2 of the Supplement.

Figure 5 shows the vertical profiles of $O_3$ production and loss terms. All calculations were performed using model data (justified by the findings from Section 3.1) as a full set of in situ observations is only available for HOOVER, but not for UTOPIHAN and BLUESKY. Figure 5a presents net ozone production rates, which were highest at the surface with values between 1 and 2 ppbv $h^{-1}$, but had large atmospheric variabilities, represented by the $1\,\sigma$ variability shades from the vertical 280   bin averaging. NOPRs then decreased with increasing altitude. For the HOOVER campaigns, $O_3$ loss dominated between 3 and 6 km altitude with NOPRs of $-58.9 \pm 73.4$ pptv $h^{-1}$. Negative NOPRs were also found for BLUESKY between 4 and 7 km with $-18.7 \pm 12.9$ pptv $h^{-1}$ and for UTOPIHAN as well as the no-lockdown BLUESKY scenario between 5 and 6 km. NOPRs were mostly positive and constant aloft. Above 10 km altitude, NOPRs were $91.7 \pm 260.9$ pptv $h^{-1}$ for UTOPIHAN (51 data points), $71.2 \pm 151.5$ pptv $h^{-1}$ for HOOVER (25 data points), $60.7 \pm 39.7$ pptv $h^{-1}$ for BLUESKY (130 data points)

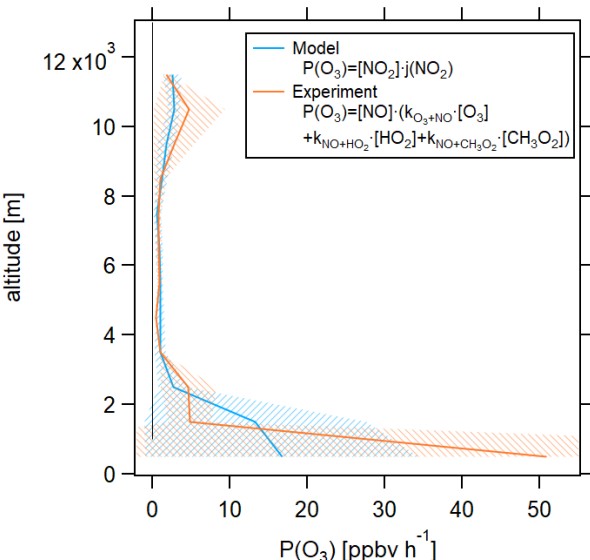

**Figure 6.** Modeled and experimental vertical profiles of P(O$_3$) for HOOVER. Modeled P(O$_3$) was calculated via NO$_2$ photolysis and experimental P(O$_3$) was calculated via the extended Leighton ratio as shown in Equation (2).

and $61.4 \pm 99.8$ pptv h$^{-1}$ for the no-lockdown scenario. The error ranges are large and overlapping and therefore, significant differences between the campaigns cannot be observed.

Figure 5b shows O$_3$ production. We calculated the P(O$_3$) via the photolysis of NO$_2$. In contrast, NO$_2$ is not available experimentally for the HOOVER campaign in which case the approximation via the extended Leighton ratio as shown in Equation (2) is necessary. Modeled P(O$_3$) via NO$_2$ photolysis and measured P(O$_3$) via reaction of NO with O$_3$, OH and HO$_2$ are in good agreement which we show in Figure 6. The only relevant deviation is observed at ground levels, where the experimental value is significantly higher compared to the modeled value. However, only three data points were available for the calculation with a $1\sigma$ standard deviation of the averaging of $>100\%$. Similar to NOPRs in Figure 5a, ground-level P(O$_3$) shows large variability with absolute values of around 10 ppbv h$^{-1}$ for BLUESKY and values of around 20 ppbv h$^{-1}$ for UTOPIHAN and HOOVER. The production term then decreased with altitude for each campaign. Significant differences between the campaigns can only be observed at high altitudes. Above 10 km, P(O$_3$) was $4.55 \pm 3.82$ ppbv h$^{-1}$ for UTOPIHAN (51 data points) and $2.68 \pm 0.90$ ppbv h$^{-1}$ for HOOVER (25 data points). For BLUESKY with the no-lockdown scenario, P(O$_3$) was $2.17 \pm 0.95$ ppbv h$^{-1}$ (130 data points) and in comparison, lockdown values were on average $0.97 \pm 0.41$ ppbv h$^{-1}$ which corresponds to a 55 % reduction in ozone production. We observed the same relative reduction as for NO and NO$_2$ mixing ratios.

Figure 5c presents the vertical profiles of O$_3$ loss via the reaction with NO which show a similar course compared to the P(O$_3$) profiles. Above 10 km, O$_3$ loss via reaction with NO was largest for UTOPIAN with $4.37 \pm 3.82$ ppbv h$^{-1}$, followed by HOOVER with $2.56 \pm 0.87$ ppbv h$^{-1}$. For BLUESKY, a loss of $0.86 \pm 0.42$ ppbv h$^{-1}$ was observed during the lockdown and

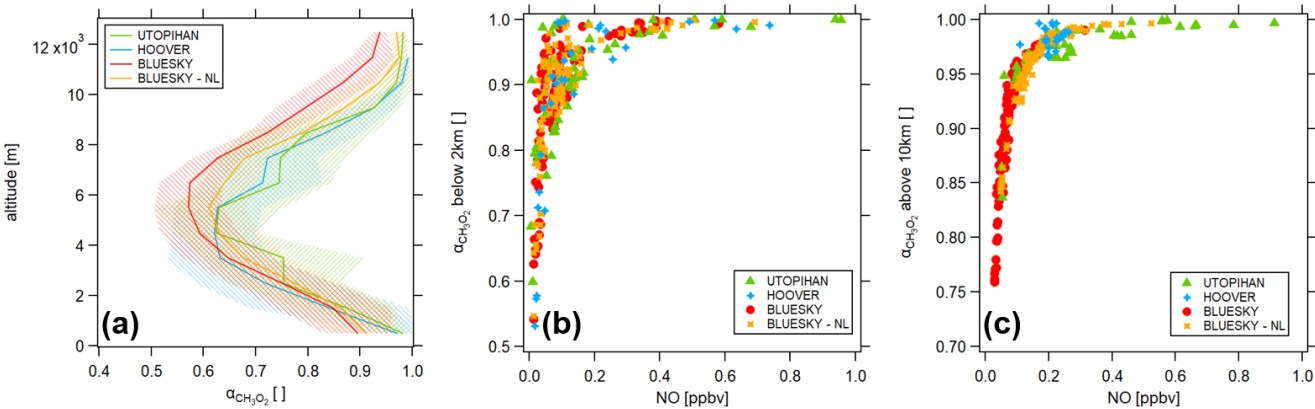

**Figure 7.** $\alpha_{CH_3O_2}$ for the campaigns UTOPIHAN (green), HOOVER (blue), BLUESKY (red) and the no-lockdown (NL) scenario (yellow) (a) as vertical profile, (b) in correlation with NO below 2 km and (c) in correlation with NO above 10 km.

a loss of $2.05 \pm 1.02 \, \text{ppbv h}^{-1}$ for the no-lockdown scenario. Figures 5d-f present additional considered loss pathways for $O_3$ via photolysis and via the reactions with $HO_2$ and OH. It can be seen that these $O_3$ losses are negligibly small in comparison to the loss via NO and no significant differences between the campaigns were present.

Consequently, net production of ozone was dominated by $NO_x$ chemistry for all campaigns and variations in production and loss terms corresponded to the mixing ratios of NO and $NO_2$ as presented in Figure 3. In the campaign comparison, higher $NO_x$ concentrations (as for example for UTOPIHAN) lead to higher production and loss terms of $O_3$ and vice versa. For the BLUESKY campaign, this analysis shows that the lockdown did not affect net ozone production rates, but instead impacted the cycling of $O_3$ such that both production and loss rates were decreased through the reduced availability of NO and $NO_2$ in the upper troposphere.

### 3.3 Chemical regime

As described above, the share of methyl peroxyradicals forming formaldeyhde $\alpha_{CH_3O_2}$ can be a measure for the dominant chemical regime when correlated with NO mixing ratios. We have previously validated this method in a comparison to the established method of analyzing the HCHO/$NO_2$ ratio (Nussbaumer et al., 2021). HCHO can be formed by almost any hydrocarbon and is therefore a proxy for VOCs which are often not measured in their entirety. Likewise, $\alpha_{CH_3O_2}$ - representing the HCHO yield from methyl peroxy radicals - is capable of revealing the dominant chemical regime without the knowledge of ambient VOC levels. Figure 7a shows the vertical profiles of $\alpha_{CH_3O_2}$ for all available data point for all campaigns based on the model simulation. $\alpha_{CH_3O_2}$ values were close to 1 at the surface and decreased with altitude up to around 5 km where values of around 0.6 were observed, with no significant differences between the campaigns. $\alpha_{CH_3O_2}$ increased again aloft whereas it was lowest for the BLUESKY campaign. Above 10 km, $\alpha_{CH_3O_2}$ was $0.97 \pm 0.03$ for UTOPIHAN, $0.98 \pm 0.01$ for HOOVER and $0.96 \pm 0.04$ for the no-lockdown scenario for BLUESKY. In comparison, $\alpha_{CH_3O_2}$ was lower for BLUESKY with $0.90 \pm 0.06$.

Figures 7b and c present $\alpha_{CH_3O_2}$ in correlation with NO mixing ratios below 2 km altitude and above 10 km altitude, respectively, based on model results. Below 2 km altitude, $\alpha_{CH_3O_2}$ ranged between 0.5 and 1.0 over the NO range of 0 - 1 ppbv.

No significant trends or differences can be observed. We show $\alpha_{CH_3O_2}$ between 2 and 10 km altitude in Figure S8 of the Supplement which does not present any differences between the campaigns either. In contrast above 10 km altitude, tropospheric $\alpha_{CH_3O_2}$ showed a different behavior for each campaign. For an easier distinction, we show each campaign in an individual panel in Figure S9 of the Supplement. For UTOPIHAN, $\alpha_{CH_3O_2}$ was high and almost non-responsive to changing NO mixing ratios with a slope of $\Delta\alpha\,/\,\Delta\text{NO} = 0.09 \pm 0.02$ ppbv$^{-1}$. In contrast, $\alpha_{CH_3O_2}$ for BLUESKY was between

0.75 and 1. Small changes in NO mixing ratios caused large changes in $\alpha_{CH_3O_2}$ with a slope of $1.12 \pm 0.08$ ppbv$^{-1}$. For the no-lockdown scenario the response of $\alpha_{CH_3O_2}$ to NO was intermediate between UTOPIHAN and BLUESKY with a slope of $0.37 \pm 0.03$ ppbv$^{-1}$. These observations suggest that a VOC limited chemical regime was present during the UTOPIHAN campaign in the upper troposphere and a transition regime during the BLUESKY no-lockdown scenario, likely due to emission control over time. For BLUESKY, we observe a distinct $NO_x$ limitation in the upper troposphere which is related to the lock-

down conditions. Aircraft $NO_x$ emissions are much larger than aircraft VOC emissions (Schumann, 2002). We can therefore expect reduced air traffic to effect lower $NO_x$/VOC ratios shifting chemistry towards a $NO_x$ limited regime. Lamprecht et al. (2021) reported ground-level reductions of several aromatic VOCs during the COVID-19 lockdown to be comparable to $NO_x$ reductions in Europe, implicating a steady $NO_x$/VOC level and therefore no changes in the dominating chemical regime, which is in line with our findings for the lower troposphere. Only few data points were available for HOOVER which were observed

at similar NO levels and the response of $\alpha_{CH_3O_2}$ to NO can therefore not be investigated. While the NOPR did not change under lockdown conditions due to compensating effects in the $NO_x$ chemistry, we can expect impacts on tropospheric ozone from changes in VOCs (including $CH_4$) relevant for future emission scenarios. The effects of $NO_x$ aircraft emissions on $O_3$ and $CH_4$ have been previously discussed for pre-lockdown conditions in Khodayari et al. (2014) and Khodayari et al. (2015) who present increased methane loss rates and a shorter lifetime as a response to increased OH concentrations from aviation

as well as higher ozone production rates. Having investigated NOPR in this study, lockdown effects on $CH_4$ loss in the UT induced by reduced air traffic could be subject to future studies.

## 4 Conclusions

In this study, we have presented in situ observations of atmospheric trace gases and model simulations from the EMAC model for three different aircraft campaigns across Europe, the UTOPIHAN campaigns in 2003/04, the HOOVER campaigns in

2006/07 and the BLUESKY campaign in 2020, including a modeled "no-lockdown scenario" with business as usual emissions for the latter. We found that model results can reproduce in situ observations well and thus could be used for further analysis which benefits from a more complete set of parameters and a higher data coverage. While observations for $O_3$, CO and $HO_x$ were very similar for all campaigns, $NO_x$ showed significant differences, particularly in the upper troposphere, where mixing ratios were highest for UTOPIHAN and HOOVER, followed by the no-lockdown scenario for BLUESKY. Observed NO

and $NO_2$ emissions during the BLUESKY campaign were approximately 55 % lower compared to the modeled no-lockdown

scenario which are attributed to reduced aircraft activity at these altitudes due to the COVID-19 travel restrictions. We found a similar trend in production and loss terms of $O_3$ which were dominated by $NO_x$ chemistry. The COVID-19 lockdown caused a significant deceleration in $O_3$ cycling whereas net ozone production rates were not affected by the emission reductions. Finally, we showed that chemistry in the upper troposphere was VOC limited during the UTOPIHAN campaign, $NO_x$ limited during

the BLUESKY campaign and in a transition regime for the BLUESKY no-lockdown scenario. While ground-level chemistry regimes were not found to be affected, the COVID-19 lockdown caused the predominant chemistry to shift from a transition regime to a clear $NO_x$ limited regime at high altitudes.

We found that the three aircraft campaigns, performed over a period of 17 years, represent the range from VOC to $NO_x$ limited tropospheric ozone chemistry, which can help analyze the impacts of anthropogenic emission scenarios. We encourage

future studies to investigate the dominating chemical regime in the upper troposphere, a topic which has not received much attention in literature so far, in order to get a deeper understanding of photochemical processes and the dominant ozone chemistry in a range of the atmosphere which receives its main $NO_x$ emissions from air traffic and lightning. The COVID-19 lockdown has been a unique opportunity to examine the effect of sharp reductions in primary pollutants on our atmosphere and could be a guidepost for future air policy in an effort to decrease anthropogenic emissions and to decelerate global warming.

*Data availability.* Data measured during the flight campaigns BLUESKY, UTOPIHAN and HOOVER are available upon request at https://keeper.mpdl.mpg.de/ to all scientists agreeing to the respective data protocols. The model results used in this study are available upon request to the author.

*Author contributions.* CMN and HF had the idea and designed the study. CMN analyzed the data and wrote the manuscript. AP provided the modeling data. IT provided NO data for BLUESKY. CO data for BLUESKY were received from LR. $O_3$ data for BLUESKY were obtained

from FO. HH provided $HO_x$ data for HOOVER. JL and HF were significantly involved in planning and operating the research campaign.

*Competing interests.* Andrea Pozzer is a member of the editorial board of Atmospheric Chemistry and Physics.

*Acknowledgements.* We acknowledge Simon Reifenberg for preparing the input data for EMAC. This work was supported by the Max Planck Graduate Center (MPGC) with the Johannes Gutenberg-Universität Mainz.

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
