# Peer review of "Tropospheric ozone production and chemical regime analysis during the COVID-19 lockdown over Europe"

_Atmospheric Chemistry and Physics, 2021_

## Author Comment (AC1)

**Referee 1:**

*The manuscript discusses aircraft observations from three campaigns over Europe. One campaign was conducted in May/June 2020, when lockdown induced emission changes provided a unique opportunity to study changes in atmospheric chemistry. The study shows that emission changes of NOx had a profound impact on tropospheric ozone production regimes. The paper is well-written, but would benefit from further detail regarding the methodology and simplifications that represent the basis of the analysis. In this context, the fact that no accompanying VOC measurements were conducted presents a shortcoming that should be addressed in more detail.*

We would like to thank the referee for the helpful feedback and the time to review our manuscript.

*Detailed comments:*

*Introduction:*

*The introduction would benefit from a more robust literature review on early papers that have unravelled the relationship between NOx and O3 chemistry.*

We have added some text with additional citations from earlier studies regarding the NOx and O3 relationship.

Lines 40 ff.: Earlier studies on evaluating correlations of NOx and O3 in the troposphere include Liu et al. (1987), Logan et al. (1985) and Lin et al. (1988), reporting a non-linear dependence that varies with ambient levels of hydrocarbons and NOx.

*Eq. 1: Why can RO2 cross reactions be ignored here? A rational should be given why these terms are omitted for the lower continental atmosphere (e.g. <4km). CH3O2 is exclusively produced from methane oxidation, however even in the remote atmosphere, methane and VOC OH-reactivity are comparable (see: Mao et al., ACP, 2009. doi: 10.5194/acp-9-163-2009). The presented simplification might work for the remote marine atmosphere, but I doubt that it is applicable to polluted continental areas in a quantitative sense as analysed here.*

We have exemplarily taken the cross reaction of CH3O2 with CH3O2 as surrogate for RO2 into account. Self-reaction of CH3O2 leads to a CH3O2 loss forming HCHO. The impact can be estimated using modeled CH3O2 or calculated CH3O2 via Equation 3 and the IUPAC rate constant k(CH3O2+CH3O2)=1.03e-13*exp(365/TK). Our calculations show that the contribution of the self-reaction of CH3O2 to CH3O2 loss (and therefore to alpha as shown in Equation 1) is negligible with on average 0.4% for BLUESKY, 0.3% for HOOVER and 0.2% for UTOPIHAN. We have added text to the manuscript to point this out.

Line 53 f.: Self-reaction of CH3O2 as a contributor to CH3O2 loss forming HCHO is negligible.

*Line 73: The authors mention emission reduction studies were only performed at the surface, but none of the cited papers actually investigated emission reductions. The cited studies investigated changes in ambient concentrations, which are typically subject to chemistry and meteorological / climatological differences. It is somewhat unclear what the authors try to say here. Aren't pollutants (with a few exceptions) primarily released at*

*the surface – or do the authors rather want to refer to the impact of emissions on atmospheric chemistry? If the authors specifically mean that emissions released above the surface (e.g. from air traffic) are of importance, I would suggest to reword this paragraph and be more specific about this.*

The cited papers report changes in pollutant concentrations during the COVID-19 lockdown. Whereas some studies find that changes in O3 mixing ratios can be (partly) attributed to meteorological changes – which we have pointed out in our manuscript – the observed changes in the abundance of primary pollutants such as NOx are not induced by meteorology and decreases are related to the lockdown regulations. To name some examples:

- Menut et al. (2020) Lines 6 ff. of Abstract:

  *"This study shows that the lockdown effect on atmospheric composition, in particular through massive traffic reductions, has been important for several short-lived atmospheric trace species, with a large reduction in NO2 concentrations, a lower reduction in Particulate Matter (PM) concentrations and a mitigated effect on ozone concentrations due to non-linear chemical effects."*

- Ordéñez et al. (2020) Lines 8 ff. of Abstract:

  *"(…) shows that the low NO2 concentrations were mostly attributed to the emission reductions while O3 anomalies were dominated by the meteorology."*

- Chossière et al. (2021) Lines 3 ff. of Abstract:

  *"Using global satellite observations and ground measurements from 36 countries in Europe, North America, and East Asia, we find that lockdowns led to reductions in $NO_2$ concentrations globally (…)."*

Most of the cited studies refer to air quality changes at the surface due to emission reductions from lockdown regulations, e.g. reduced traffic, and only very few studies consider lockdown induced changes in the free troposphere, which is why we believe our study can be an important contribution to the literature.

We have rephrased the sentence in the manuscript and now mention pollutant reductions (which is observed) instead of emission reductions (which is the origin) to avoid confusion and to emphasize that we are talking about mixing ratios, and not fluxes.

Line 75 ff.: Most of the literature on pollutant reductions during the COVID-19 lockdown focuses on near-surface air quality and only few studies consider the free troposphere.

*Line 100: ok here methyl peroxy radicals are simply based on methane, but earlier (line 47), methanol was also mentioned as an important precursor for a study site in Finland. In fact there could be many more precursor VOCs for methyl peroxy radicals in the upper atmosphere (e.g. the photolysis of carbonyls, or subsequent RO2 x HO2 and RO2 x RO2 reactions of most carbonyls)*

Thank you for pointing towards methanol. In fact, CH3O2 is formed from acetaldehyde and not from methanol. We have corrected this in the manuscript.

Line 49 ff.: CH3O2 formed from e.g. the oxidation of acetaldehyde (CH3CHO) or methane (CH4) …

We would like to point out that calculating CH3O2 via Equation 3 was originally derived by Bozem et al. (2017), doi: 10.5194/acp-17-10565-2017 and comparing with modeled CH3O2 shows good agreement in the upper troposphere (Figure 2g). Additionally, we have calculated ozone production via the photolysis of NO2 from model simulations and via the extended Leighton ratio (Eq. 2) using experimental data for the HOOVER campaign and find good agreement. We therefore believe this calculation to provide a reasonable estimate. We have added the vertical profiles showing the comparison of modeled and experimental NOPR for HOOVER as Figure 6 to the manuscript.

Lines 289 ff.: Modeled P(O3) via NO2 photolysis and measured P(O3) via reaction of NO with O3, OH and HO2 are in good agreement which we show in Figure 6. The only relevant deviation is observed at ground levels, where the experimental value is significantly higher compared to the modeled value. However, only three data points were available for the calculation with a 1σ standard deviation of the averaging of >100%.

*Using a campaign in Africa as a reference seems a stretch here. What about the role of biogenic VOCs and oxidation products? For example Crutzen et al., (Atmos. Environ., 2000: doi: 10.1016/S1352-2310(99)00482-3) found significant amounts of BVOCs and their oxidation products in the tropics up to 10 km.*
*What is the bias of neglecting other RO2 sources (e.g. changes in anthropogenic VOCs and BVOCs) over Europe? Also, May / June represent seasons where biogenic emissions in Europe should play an increasing role.*

We agree with the referee and have investigated the proportion of CH3O2 in the overall modeled RO2 across Europe.

[Figure]

This figure shows the ratio of modeled CH3O2 and RO2 as a function of the pressure level for the BLUESKY campaign. It can be seen that above 800hPa, CH3O2 represent > 90 % of the overall modeled RO2. In the lower troposphere CH3O2 accounts for on

average ~ 70 % of RO2. We have added the figure to the Supplement and some text to the manuscript.

Lines 104 ff.: We assume RzO2 (the sum of all peroxy radicals) to be represented by CH3O2 which we find to be a reasonable approximation when comparing modeled CH3O2 to the overall modeled RO2 as shown in Figure S1 of the Supplement, exemplarily for the BLUESKY campaign. Above 800 hPa, CH3O2 represents more than 90 % of RO2. Below 800 hPa, it still accounts for more than 70 % on average.

We would additionally like to point out that this estimation of CH3O2 is only used when calculating O3 production from the measured mixing ratios. For the most part of our analysis, we use the photolysis of modeled NO2 in order to estimate ozone production. We have performed a sensitivity study of ozone production for HOOVER regarding RO2 which can be seen in the following figure. The blue profile shows ozone production when RO2 is equal to CH3O2 and the orange profile shows a case where CH3O2 represents around a third of the overall RO2. Small differences can be observed in the lower troposphere which are not significant. The error on P(O3) resulting from approximating RO2 through CH3O2 is therefore negligible.

[Figure]

Line 145: So the campaign was conducted in May/June, when most lockdown related restrictions were already easing in Europe – would the analysis presented here then be more of a reflection of the post-lockdown regime, with some restrictions (e.g. travel restrictions) still in place, others not? For example traffic volumes across Europe and elsewhere (e.g. China) largely recovered by June 2020.

We agree with the referee that most restrictions across Europe were in place in March and April. The campaign was conducted from May 21 to June 9, which presents a time period during which emissions increased again due to for example higher traffic loads, but have not got back to the original pre-lockdown level, e.g. the Covid-19 Mobility Project (http://covid-19-mobility.org/) reported a mobility reduction of still more than 10% compared to 2019 during this time interval in Germany. This is in line with our pollutant reduction observations: we found a decrease in NOx mixing ratios at the surface which was smaller compared to findings from other studies focusing on data from March and April (Lines 218 ff. of the manuscript). We have added some text in the manuscript for clarification.

Lines 158 ff.: While most restrictions across Europe were in place in March and April 2020, May and June emissions, particularly from air travel but also from ground-based sources such as onroad traffic, were still affected by the COVID-19 lockdowns (Schlossel et al., 2020; Brockmann Lab, 2022; Hasegawa, 2022; EUROCONTROL, 2022).

*Line 144 ff: No VOCs were measured during these campaigns, which presents a major uncertainty. At the minimum the authors should state something about anthropogenic VOC emission changes and estimate the potential change in VOC reactivity prior and after the lockdown relative to NOx. Observations of lockdown induced reductions of anthropogenic VOCs are sparse. In Europe there is evidence that reductions were significant, comparable to NOx (see Lamprecht et al., ACP, 2020: doi: 10.5194/acp-21-3091-2021).*

We do not believe the lack of VOCs to be a drawback of our study. The introduction of the measure $\alpha CH_3O_2$ replaces the need of VOC measurements, particularly in the upper troposphere where $CH_3O_2$ is around 90% of overall $RO_2$ (see above). However, we appreciate the literature suggestion of the referee and have added some discussion on lockdown induced VOC changes. Aircraft $NO_x$ emissions are much larger than aircraft VOC emissions (Schumann, 2002). We therefore expect a much larger decrease in $NO_x$ than VOC in the UT, which would shift the $NO_x$/VOC ratio to smaller values and towards a $NO_x$ limited chemical regime which is line with our observations. Lamprecht et al. (2020) find that VOC reductions at the surface are similar to $NO_x$ reductions which implicates an unchanged $NO_x$/VOC ratio and corresponds to what we report for the lower troposphere.

Lines 335 ff.: Aircraft $NO_x$ emissions are much larger than aircraft VOC emissions (Schumann, 2002). We can therefore expect reduced air traffic to effect lower $NO_x$/VOC ratios shifting chemistry towards a $NO_x$ limited regime. Lamprecht et al. (2021) reported ground-level reductions of several aromatic VOCs during the COVID-19 lockdown to be comparable to $NO_x$ reductions in Europe, implicating a steady $NO_x$/VOC level and therefore no changes in the dominating chemical regime, which is in line with our findings for the lower troposphere.

*Line 168: Comment on: "the model is generally capable of reproducing the experimental data": Looking at the ozone profile, it does not seem that the 3D model has a very robust predictive capability for ozone. In fact, Figure 2b shows a model offset between 10-20 ppbv for ozone concentrations around 50 – 60 ppbv (e.g. mid – troposphere), which is significant for ozone! For example regional AQ models typically reproduce tropospheric ozone within 5 ppbv when ozone concentrations are around 60 ppbv (e.g. Im et al., Atm. Environ., 2015; doi: 10.1016/j.atmosenv.2015.02.034 ). These AQ CTMS show biases at the high (e.g. >90ppbv) and low end (<30 ppbv), but not so much in the range of 50-60 ppbv. Is there an explanation for the large model bias in the mid to upper troposphere? Both $HO_2$ and $CH_3O_2$ are overpredicted in the mid troposphere – it appears that additional $RO_x$ losses are missing in the model, or that the simplified experimental analysis for $CH_3O_2$ has limitations. Could the representation of clouds and liquid chemistry be a limitation, or are additional losses of $RO_2$ x $RO_2$/$HO_2$ type reactions missing?*
*There is also indirect evidence of uncertain (e.g. VOC?) chemistry. For example, in Figure 2 g a comparison between modelled and experimental $CH_3O_2$ concentration is shown. The mean difference from 3 km upward can be as large as a factor of 2! For comparison measurements and modelling by Ridley et al. (JGR, 1992, doi: 10.1029/91JD02287) showed that in the remote marine free troposphere, where $CH_4$*

*and CO dominate, peroxy radicals estimated from a photo stationary state assumption can be largely reconciled with a photochemical box model.*

The ozone bias, the referee is pointing out, is something almost all global models suffer from and the causes of which have not been conclusively understood so far. Revell et al. (2018) found a positive O3 bias in the northern hemisphere for the models of the CCMI (Chemistry-Climate Model Initiative). Other studies reported similar observations for global models (Young et al. (2018), Jöckel et al. (2016), Parrish et al. (2014)). Regional models can be tuned to match O3 observations more easily, but lack the complexity of global models which need to encompass the entire globe. We find a model O3 overestimation by <20% for HOOVER and BLUESKY and by <10% for UTOPIHAN on average. Given the observations of Revell et al. (2018) that the O3 bias can be as large as 40-50% in the northern hemisphere, we believe our results to provide estimates at the more reasonable end. Regarding CH3O2, Figure 2g shows close agreement for calculated and modeled CH3O2 for the lower and upper troposphere which is the area which we predominantly study in this work. We have added some text to the manuscript pointing towards the O3 bias for global models.

Lines 196 ff.: Model values were approximately 20% higher compared to the measured data, but showed the same vertical shape. The observed positive O3 bias of the modeled data is an issue almost all global models suffer from in the northern hemisphere and which has not been entirely understood yet (Revell et al., 2018; Young et al., 2018; Jöckel et al., 2016; Parrish et al., 2014).

*Section 3.2*

*Line 212. Considering that the presented analysis in this section is exclusively based on the ECHAM/MESSy model scenarios here, why leave this statement as a possibility or "possible explanation"?  It should be pretty straight forward to get the emissions data from the model and compare the model projected changes quantitatively. E.g. how much have NOx emissions in the model then changed between 2003 and 2021?*

We agree with the referee. We have added a Figure to the Supplement showing how NOx emissions in the model decreased over the past 20 years and rephrased our statement in the main text.

[Figure]

NOx emissions from Europe (Tg(N)/yr)

Lines 227 ff.: The differences in NO mixing ratios between the campaigns are the outcome of the general emission reduction due to legislative limitation of nitrogen oxides and other hazardous pollutants over the past decades, as the campaigns took place 15-20 years apart. We show the decrease in NOx emissions in the model over the past two decades in Figure S5 of the Supplement.

*Section 3.3*

*While I understand that alpha could be a semiquantitative experimental measure for investigating the chemical regime of ozone production, I wonder whether this section represents a little bit of a circular argument: since the analysis is largely based on the output of a chemical Earth system model (ECHAM/MESSy) anyway, why not also use established methods (e.g. Kleinman et al., GRL, 1997; doi: 10.1029/97GL02279) to investigate net P(O3) changes prior and post lockdown. From Figure 5 b and 5c, the relationship between alpha and NO for the individual campaigns does not seem to be dramatically different. From Figure 5a the difference between BLUESKY and BLUESKY-NL seems to be smaller than the uncertainty of both. So how robust are the findings? For example, if lockdown induced changes in anthropogenic VOC and NOx are proportional, one would expect to move sideways down along ozone isopleths. In this context it would be interesting to calculate the OH reactivity from the model. From the presented results and analysis, I have the impression that it is assumed to be dominated by CH4 and CO. While perhaps plausible in the upper remote atmosphere, it is hard to believe that VOCs wouldn't play a significant role in the lower 3-5 km. Even in the remote (marine/coastal) atmosphere (e.g. Mao et al., ACP, 2009. doi: 10.5194/acp-9-163-2009) observations show that the VOC reactivity accounts for 20%. Aircraft studies have shown that models significantly underpredict VOC reactivity above North America (e.g. Chen et al., ACP, 2019: doi: 10.5194/acp-19-9097-2019), and that the VOC reactivity likely accounts for more than 40-50% in the FT over continental areas. This has been shown by many aircraft studies (e.g. Schroeder et al., Elementa, 2020: doi: 10.1525/elementa.400; Hu et al., JGR, 2014: doi: 10.1002/2014JD022627).*

We would like to point out, that this study is originally based on in-situ observations during the three field campaigns UTOPIHAN, HOOVER and BLUESKY. For consistency,

we are using the model output for the net ozone production calculations as some of the trace gas were not measured across all three campaigns, e.g. OH or HO2. The HOOVER campaign provides a full dataset of all trace gas measurements which we use as validation of the model. To our understanding the method presented by Kleinman et al. (1997) requires the knowledge on hydrocarbon concentrations which were not measured during the field campaigns. The advantage of using the described alpha value is that VOC measurements are not necessary in order to determine the dominant chemical regime. We have previously compared this method in Nussbaumer et al. (2021, doi: 10.5194/acp-21-18413-2021) to the established method of the HCHO to NO2 ratio (Sillman, 1995; Duncan et al., 2010; Martin et al., 2004) and found good agreement. We therefore believe that alpha is a powerful tool regarding the determination of dominant ozone chemistry in the troposphere.

We agree with the referee that no significant differences between the campaigns, including the BLUESKY and the BLUESKY-NL scenario, can be observed for Figure 7b (5b previously) in the lower troposphere. In contrast, for the upper troposphere in Figure 7c we do observe major differences between the campaigns. The UTOPIHAN alpha values are almost non-responsive to changing NO, reflected in the slope of $0.09 \pm 0.02$ ppbv$^{-1}$. In contrast, the BLUESKY alpha values change rapidly for small NO changes with a slope of $1.12 \pm 0.08$ ppbv$^{-1}$. Intermediate results are obtained for the BLUESKY-NL alpha values, showing a slope of $0.37 \pm 0.03$ ppbv$^{-1}$. These differences are mathematically significant and provide strong evidence on a changing regime in the upper troposphere in contrast to the lower troposphere where we do not make these observations. This is also shown in Figure 7a. Above ~8km the difference between the BLUESKY and the BLUESKY-NL alpha values is larger than the individual uncertainties and above ~10km even larger than the combined uncertainty. In the lower troposphere - we agree with referee – no relevant differences are observed which strongly underlines our conclusion.

We thank the referee for the suggestion of calculating OH reactivities from the model. However, we believe this to be outside the scope of this study. While VOCs and NOx are precursors to Ozone formation, we do not require the full knowledge of both in order to determine the dominant chemical regime. An example for this is the HCHO/NO2 ratio, a method well established in current literature. As HCHO can be formed by almost any hydrocarbon it can be seen as a surrogate for VOCs. Similarly, the alpha value in this study representing the HCHO yield from methyl peroxy radicals is capable of revealing the dominant chemical regime without a full set of VOC measurements.

We have added some text in the manuscript for clarification.

Lines 314 ff.: We have previously validated this method in a comparison to the established method of analyzing the HCHO/NO2 ratio (Nussbaumer et al. (2021)). HCHO can be formed by almost any hydrocarbon and is therefore a proxy for VOCs which are often not measured in their entirety. Likewise, $\alpha CH3O2$ - representing the HCHO yield from methyl peroxy radicals - is capable of revealing the dominant chemical regime without the knowledge of ambient VOC levels.

*In summary: Putting the analysis more into context of the above mentioned literature and performing some sensitivity analysis on VOC reactivity would help clarify uncertainties that are associated with the main findings on ozone sensitivity.*

*Minor comments:*

*Line 30: Reaction R1 has already been described by Leighton*

Thank you, we have added this reference to the manuscript.

Line 30 ff.: Under the influence of sunlight, NO2 can subsequently form O3 through the reaction with molecular oxygen as shown in Reaction (R1) (Leighton, 1961; Crutzen, 1988; …).

*Line 35: This has already been shown by many studies in the 70ies and early 80ies (e.g. Calvert and Stockwell, Can. J. Chem., 61, 1983).*

We have added the suggested reference.

Line 35 ff.: (…), which means that a reduction in ambient NOx can either increase or decrease O3 production (Calvert and Stockwell, 1983; Pusede et al., 2015).

*Line 44: what is meant by share here? the authors seem to refer to a fraction or a ratio in eq. (1)*

Yes, the referee is correct. We have changed the word 'share' to 'fraction'.

Line 46 ff.: We have recently shown that the fraction $\alpha$ of methyl peroxyradicals (CH3O2) forming formaldehyde (HCHO) in correlation with ambient NO concentrations (…).

*Line 300: The authors sometimes put units in brackets e.g. [ppbv], but for alpha, which is a relative quantity, an empty bracket [ ] seems somewhat arbitrary.*

We meant to indicate with the empty bracket that alpha is unitless. However, we have deleted the empty brackets in the main text in order to avoid confusion.

Line 329: … $\Delta\alpha \, / \, \Delta NO = 0.09 \pm 0.02 \, \text{ppbv}^{-1}$…

---

## Author Comment (AC2)

**Referee 2:**

*This paper reports on new and exciting data from the BLUESKY experiment conducted ,during the early months of the COVID-19 pandemic. The paper is well written and the figures are clear and appropriate. In general I think that the aim of the paper is quite novel, and that the results could be useful for the research community. However, I have reservations about the analysis because the BLUESKY ozone data in 2020 are not lower than the ozone values in the earlier campaigns. In contrast, three new studies show ozone above Europe was anomalously low in 2020. This discrepancy needs to be reconciled before I can recommend the paper for publication.*

We would like to thank the referee for the helpful feedback and the time to review our manuscript.

*Major comments:*

*1) Three recent papers have shown a clear decrease of free tropospheric ozone above Europe during 2020, in association with the COVID-19 economic downturn. Two of these papers have been cited [Steinbrecht et al., 2021; Clark et al., 2021]. The third paper is by Chang et al., 2022, and it will appear any day as an accepted paper in AGU Advances (it will be posted here: https://agupubs.onlinelibrary.wiley.com/toc/2576604x/0/ja ).*

We have added this reference to our manuscript.

Lines 77 ff.: Steinbrecht et al. (2021), Chang et al. (2022) and Bouarar et al. (2021) reported decreases in O3 concentrations in the free troposphere based on in situ observations and modeling studies in the northern hemisphere.

*Given that 2020 was an anomalously low year for ozone, it is very puzzling as to why the BLUESKY ozone observations are not lower than the UTOPIHAN or HOOVER data. One possibility is that the sample size of these datasets is too low to provide an accurate estimate of monthly or seasonal mean ozone. Three papers have looked at the sample size necessary to quantify monthly mean ozone above Europe and determined that 12-20 profiles are necessary [Logan et al., 1999; Saunois et al., 2012; Chang et al., 2020]. Given that the IAGOS program has dozens of profiles per month from Frankfurt, you could compare your monthly mean profiles to those from IAGOS. The IAGOS monthly means will be accurate due to the high number of profiles and you can then determine if the aircraft campaign data are biased high or low.*

Thank you for this helpful suggestion. We have analyzed the IAGOS ascending and descending profiles over Frankfurt during the BLUESKY campaign and found a very good agreement with our measured O3. Please note that the measured O3 data were filtered for the modeled tropopause pressure and the IAGOS data were filtered for 250ppbv O3 to match the vertical profile above ~10km.

[Figure]

The BLUESKY campaign took place in May/June 2020. In comparison, the HOOVER campaign was partly carried out in October 2006 and in July 2007. Similarly, the UTOPIHAN campaign took place in June/July 2003 and March 2004. Due to the seasonality of ozone, being highest during summer months and lowest during winter months, the presented graphs do not show the lower O3 values in 2020 described in the studies mentioned by the referee. The following graphs show the vertical profiles of modeled O3 and measured O3, separated into different seasons for the HOOVER and the UTOPIHAN campaign. When comparing just the summer data (center), we also observe lower O3 values, both for the model and the measured data.

[Figure]

We present the measured and modeled vertical O3 profiles showing the seasonality in Figure S6 of the Supplement and have added text to the manuscript for explanation.

Lines 248 ff.: While ozone concentrations are dependent on various effects such as precursor levels (including NOx and VOCs) or meteorology, seasonal variations with a maximum around summer time and a minimum during winter months are also of importance (Logan et al., 1985). The here shown campaigns include different seasons: the HOOVER campaigns took place in October and July and the UTOPIHAN campaigns include data from July and March. Figure S6 of the Supplement shows the vertical profiles of ozone separated into different seasons, for both modeled and measured data. Comparing late spring / early summer data of the three field campaigns reveals that O3 levels during BLUESKY were lower compared to HOOVER and UTOPIHAN which is in line with findings from Clark et al. (2021), Chang et al. (2022), Bouarar et al. (2021) and Miyazaki et al. (2021).

*2) Given that the three studies mentioned above report anomalously low ozone above Europe in 2020, we can conclude that net ozone production was below average in 2020, which matches the findings of Miyazaki et al., 2021. However, your conclusion seems to be that net ozone production was not unusual in 2020. How can you reconcile these different conclusions?*

We would like to emphasize that net ozone production predominantly depends on the concentrations of O3, NO2 and NO, whereas ozone production occurs via NO2 photolysis and ozone loss is mostly represented by the reaction of O3 with NO. All three species were found to be lower in 2020 and therefore the impact on net ozone production depends on whether the decline in the production or the loss term prevails. Miyazaki et al., 2021 reports decreases in O3 across all of the troposphere which is line with our findings, but to our understanding does not present any observations regarding net ozone production rates.
The goal of our study is to investigate ozone production and the chemical regime during the COVID-19 lockdown, where the central tool is the comparison of the lockdown to a modeled no-lockdown / business-as-usual scenario. We conclude that while we observe both lower ozone production and lower ozone loss during the lockdown compared to a modeled no-lockdown scenario, the net production is not impacted (and only the O3 cycling is slowed down). For drawing any conclusions on how the lockdown changed net ozone production, we prefer to avoid a direct comparison of the BLUESKY campaign with the earlier campaigns HOOVER and UTOPIHAN as many more factors such as meteorology, seasonality or year-to-year decline of pollutants due to legislation contribute to this development, which we cannot quantify.

*3) Bouarar et al., 2021 concluded:*

*"Zonally averaged ozone in the free troposphere during Northern Hemisphere spring and summer is found to be 5%–15% lower than 19-yr climatological values, in good agreement with observations. About one third of this anomaly is attributed to the reduction scenario of air traffic during the pandemic". As conclusion that the reduction of aircraft emissions impacted ozone in 2020 has already been published, you should specifically mention this finding in your paper. It would also be helpful to explain how aircraft emissions have strongly increased over the past 20 years [Lee et al., 2021].*

We agree with the referee. We have included the references and some text to the manuscript for pointing out the findings of the mentioned studies.

Lines 77 ff.: Bouarar et al. (2021) found that reduced air traffic - a unique incidence after strongly increasing aircraft activities over the past decades as shown by Lee et al. (2021) - can explain around a third of the observed O3 decrease in 2020, the remaining contributions coming from ground-level reductions and meteorological differences.

*4) I don't agree with this statement in the Conclusions:*

*"We encourage future studies to investigate governing chemistry in the upper troposphere, a topic which has not received much attention in literature so far"*

*I know of many measurement and modelling studies of the chemistry of the upper troposphere, and a few that immediately come to mind are: Barth et al., 2021; Brunner et al., 1998,2001; Cooper e t al., 2006; Huntrieser et al., 2002; Li et al., 2001,2005; Ridley et al., 1994.*

*If your comment is meant to refer to a specific chemical process in the upper troposphere, please make that point clear.*

We regret that we were unclear with this statement. We meant to refer to the investigation of the dominating chemical regime in the upper troposphere, which to our knowledge has not been comprehensively explored. We have clarified this in the text.

Lines 364 ff.: We encourage future studies to investigate the dominating chemical regime in the upper troposphere, a topic which has not received much attention in literature so far (…).

*Minor comments:*

*First line of the Abstract:   lead should be led*

Thank you, we have corrected that.

Lines 2 ff.: The COVID-19 (Coronavirus disease 2019) European lockdowns have led to a significant reduction in the emissions of primary pollutants such as NO (nitric oxide) and NO2 (nitrogen dioxide).

*Line 28-29*

*This line mentions ozone impacts on humans, animals and plants*

*"NOx directly impacts the production of tropospheric ozone (O3) which is a hazard to human, animal and plant health (Nuvolone et al., 2018)."*

*However, the reference only deals with impacts on humans.  A good reference for the impact of ozone on plants is Mills et al., 2018.  I do not know of any authoritative papers that report ozone impacts on animals. If the authors know of such a paper they need to cite it, otherwise, impacts on animals should not be stated as there seems to be no convincing evidence.*

We agree with the referee. We have added the suggested literature on ozone effects on plant health and have deleted the statement on animal health.

Lines 28 ff.: NOx directly impacts the production of tropospheric ozone (O3) which is a hazard to human and plant health (Nuvolone et al., 2018, Mills et al., 2018).

*Line 76*

*When reviewing studies that indicate ozone reduction in the free troposphere, you should also mention two recent studies that show ozone reductions at high elevation sites within the European boundary: Cristofanelli et al., 2021; WMO Air Quality and Climate Bulletin, 2021.*

We have added these references and included them in the main text.

Lines 82 ff.: Cristofanelli et al. (2021) reported lower O3 concentrations above the PBL in 2020 compared to the 1996 - 2019 average at Mount Cimone in Italy which is in line with findings by the World Meteorological Organization (2021), extended to include two mountain sites in Germany.

*Line 166*

*"what" should be "which"*

We have corrected this in the manuscript.

Lines 178 ff.: As a full set of in situ observations necessary for a regime analysis and calculating net ozone production rates, which includes (…)

*Line 192*

*"trend" is not the right word as it refers to a change with time. Would "gradient" work better?*

Yes, we agree with the referee. We have replaced the word 'trend' with 'gradient'.

Lines 207: Figure 2f shows the vertical profiles of CH4 which did not show any particular gradient with altitude.

*References*

*Barth, M.C., Cantrell, C.A., Brune, W.H., Rutledge, S.A., Crawford, J.H., Huntrieser, H., Carey, L.D., MacGorman, D., Weisman, M., Pickering, K.E. and Bruning, E., 2015. The deep convective clouds and chemistry (DC3) field campaign. Bulletin of the American Meteorological Society, 96(8), pp.1281-1309.*

*Brunner, D., J. Staehelin, and D. Jeker (1998), Large-scale nitrogen oxide plumes in the tropopause region and implications for ozone, Science, 282, 1305–1309, doi:10.1126/science.282.5392.1305.*

*Brunner, D., J. Staehelin, D. Jeker, and H. Wernli (2001), Nitrogen oxides and ozone in the tropopause region of the Northern Hemisphere: Measurements from commercial aircraft in 1995/1996 and 1997, J. Geophys. Res., 106, 27,673– 27,699, doi:10.1029/2001JD900239.*

Chang, K.-L., et al. (2020), Statistical regularization for trend detection: An integrated approach for detecting long-term trends from sparse tropospheric ozone profiles, Atmos. Chem. Phys., 20, 9915–9938, https://doi.org/10.5194/acp-20-9915-2020

Chang, K.-L., O. R. Cooper, A. Gaudel, M. Allaart, G. Ancellet, H. Clark, S. Godin-Beekmann, T. Leblanc, R. Van Malderen, P. Nédélec, I. Petropavlovskikh, W. Steinbrecht, R. Stübi, D. W. Tarasick, C. Torres (2022), Impact of the COVID-19 economic downturn on tropospheric ozone trends: an uncertainty weighted data synthesis for quantifying regional anomalies above western North America and Europe, AGU Advances, in press: https://agupubs.onlinelibrary.wiley.com/journal/2576604x

Cooper, O. R., et al. (2006), Large upper tropospheric ozone enhancements above midlatitude North America during summer: In situ evidence from the IONS and MOZAIC ozone measurement network, J. Geophys. Res., 111, D24S05, doi:10.1029/2006JD007306.

Cristofanelli, P.; Arduni, J.; Serva, F. et al. Negative Ozone Anomalies at a High Mountain Site in Northern Italy during 2020: A Possible Role of COVID-19 Lockdowns? Environ. Res. Lett. 2021, 16 (7), 074029. https://doi.org/10.1088/1748-9326/ac0b6a

Huntrieser, H., Feigl, C., Schlager, H., Schröder, F., Gerbig, C., Van Velthoven, P., Flatøy, F., Théry, C., Petzold, A., Höller, H. and Schumann, U., 2002. Airborne measurements of NOx, tracer species, and small particles during the European Lightning Nitrogen Oxides Experiment. Journal of Geophysical Research: Atmospheres, 107(D11), pp.ACH-5.

Lee, D.S., Fahey, D.W., Skowron, A., Allen, M.R., Burkhardt, U., Chen, Q., Doherty, S.J., Freeman, S., Forster, P.M., Fuglestvedt, J. and Gettelman, A., 2021. The contribution of global aviation to anthropogenic climate forcing for 2000 to 2018. Atmospheric Environment, 244, p.117834.

Li, Q., D. J. Jacob, R. Park, Y. Wang, C. L. Heald, R. Hudman, R. M. Yantosca, R. V. Martin, and M. Evans (2005), North American pollution outflow and the trapping of convectively lifted pollution by upper-level anticyclone, J. Geophys. Res., 110, D10301, doi:10.1029/2004JD005039.

Li, Q., Jacob, D.J., Logan, J.A., Bey, I., Yantosca, R.M., Liu, H., Martin, R.V., Fiore, A.M., Field, B.D., Duncan, B.N. and Thouret, V., 2001. A tropospheric ozone maximum over the Middle East. Geophysical Research Letters, 28(17), pp.3235-3238.

Logan, J. A.: An analysis of ozonesonde data for the troposphere: Recommendations for testing 3-D models and development of a gridded climatology for tropospheric ozone, J. Geophys. Res.- Atmos., 104, 16115–16149, 1999.

Mills, G., et al. (2018), Tropospheric Ozone Assessment Report: Present-day tropospheric ozone distribution and trends relevant to vegetation, Elem. Sci. Anth., 6(1):47, DOI: https://doi.org/10.1525/elementa.302

Miyazaki, K., Bowman, K., Sekiya, T., Takigawa, M., Neu, J. L., Sudo, K., Eskes, H. (2021). Global tropospheric ozone responses to reduced NOx emissions linked to the COVID-19 worldwide lockdowns. Sci. Ad., 7 (24), eabf7460. doi: 10.1126/sciadv.abf7460

Ridley, B. A., J. G.Walega, J. E. Dye, and F. E. Grahek (1994), Distributions of NO to the upper troposphere, J. Geophys. Res., 109, D17305, doi:10.1029/2004JD004769.

Saunois, M., Emmons, L., Lamarque, J.-F., Tilmes, S., Wespes, C., Thouret, V., and Schultz, M.: Impact of sampling frequency in the analysis of tropospheric ozone observations, Atmos. Chem. Phys., 12, 6757–6773, https://doi.org/10.5194/acp-12-6757-2012, 2012.

WMO Air Quality and Climate Bulletin, No. 1, September 3, 2021; Editors: O. R. Cooper, R. S. Sokhi, J. M. Nicely, G. Carmichael, A. Darmenov, P. Laj and J. Liggio; a publication of the World Meteorological Organization, https://library.wmo.int/index.php?lvl=notice_display&id=21942#.YTIzN9

---

## Author Comment (AC3)

**Lin Tan:**

*Dear Clara,*

*Many thanks for this interesting study. Below I would summarize my understanding about your work and provide some comments if you find them useful.*

*The study aims to investigate the changes in the vertical distributions of atmospheric species over Europe before and during the COVID-19 pandemic. The pre- and intra-pandemic vertical distributions were measured by 3 aircraft campaigns: UTOPIHAN campaigns in 2003/2004, the HOOVER campaigns in 2006/2007, and the BLUESKY campaign in 2020. The model ECHAM5/MESSy2 Atmospheric Chemistry (EMAC) is run in a pre-pandemic scenario (also known as the no-lockdown scenario); the model data subsampled along with the flight tracks of the three campaigns areused to compare with the observations. The model data are first validated against HOOVER and are found to reproduce the HOOVER observations, including the trends. Then, assuming that the pre-pandemic atmospheres remain the same, this study compares the intra-pandemic observations by BLUEKSY with hypothetical pre-pandemic BLUESKY measurements constructed using the model data. A major finding is that in addition to the significant drop in major pollutants at the surface that are related to car exhausts such as $NO_x$ and CO, there is also a significant drop in $NO_2$ in the upper troposphere at 10 km, which is likely due to the reduced air traffic. Nonetheless, this study finds that the production rate of $O_3$ in the upper troposphere remains unchanged despite the $NO_x$ change. Another major finding of this study is that the chemistry regimes in the upper troposphere might have changed from a VOC-limited chemistry in the pre-pandemic era to a $NO_2$-limited chemistry in the intra-pandemic era.*

Dear Lin,

Thank you very much for your feedback and your time to read our manuscript. We found your comments really helpful and have implemented your suggestions in the manuscript draft.

*I have a few minor comments and hopefully you may find them helpful:*

*If I understand it correctly, in both Figures 2 and 3, there is only one model simulation: the ECHAM5/MESSy2 Atmospheric Chemistry (EMAC) that was run in the no-lockdown scenario. But Figures 2 and 3 may give an impression that there were different simulations separately for HOOVER and BLUESKY. Similarly, calling the subsampled model data on the BLUESKY flight path as BLUESKY-NL also made me think that there was another BLUESKY campaign before the lockdown. Would something like EMAC(on HOOVER path) and EMAC(on BLUESKY path) be clearer?*

Thank you for pointing this out. Figure 2 shows the comparison of the vertical profiles for HOOVER for the measured data in orange and the modeled data along the flight track in blue. Figure 3 shows only modeled data generated from EMAC along the flightpaths of each individual campaign. Two simulations were run on the BLUESKY path: the lockdown and the no-lockdown scenario. We have tried to clarify this in the caption of the Figures.

Figure 2 Lines 3 f.: Blue colors show modeled data by EMAC along the HOOVER campaign flight track (Model) and orange colors show experimental data (Experiment).

Figure 3 Lines 2 ff.: All data shown here are from EMAC model simulations along the flight track of each research campaign. Two separate simulations were run on the flight path of BLUESKY, one with lockdown (BLUEKSY) and one with business-as-usual emissions (BLUESKY-NL).

*Since you found that there was more NOx in the upper troposphere before the pandemic, have the possible self-contamination due to the NOx emission of the aircraft itself been removed or calibrated in order to establish the robustness of the NOx decrease from the pre-pandemic era to the post-pandemic era?*

We can exclude self-contamination due to the flight trajectories. Usually self-contamination only occurs when flying narrow curves with simultaneous advection or performing return flights on the same flight trajectories, both of which were avoided during the campaign.

*Figure S4–S6 are important results of this study.  Especially, Figure S4 demonstrates the impact of air traffic in the model, which is one of the two major conclusions of this study.  I strongly think that these 3 figures should be put in the text.  The x-axis range of Figure S5 could probably be either re-adjusted or re-plotted using the log scale for better data representation.*

Thank you for this suggestion. Figure S5 (now Figure S7) shows the same vertical profiles as Figure 3d, just on a different x-axis scale. We have added some note for clarification. We have added Figures S4 and S6 and some explaining text to the main manuscript.

Figures S7: NO2 vertical profiles during UTOPIHAN, HOOVER, BLUESKY with and without lockdown scenario as in Figure 3d of the main manuscript, but showing the full x-axis range.

Lines 241 ff.: The observed NO reduction in the upper troposphere can be attributed to reduced air traffic which we show in Figure 4. In addition to the vertical profiles of NO for BLUESKY (red) and BLUESKY-NL (yellow), we present the modeled BLUESKY-NL scenario without aircraft emissions in blue. In the lower troposphere, where aircraft emissions do not play a significant role, this profile is identical to the BLUESKY-NL scenario. In the upper troposphere, it is very similar to the BLUESKY scenario (including the air travel restrictions), showing that reduced air traffic causes the observed NO decrease.

Lines 289 f.: Modeled P(O3) via NO2 photolysis and measured P(O3) via reaction of NO with O3, OH and HO2 are in good agreement which we show in Figure 6.

*The conclusion "While the NOPR did not change under lockdown conditions due to compensating effects in the NOx chemistry, we can expect impacts on tropospheric ozone from changes in VOCs (including CH4) relevant for future emission scenarios."  Maybe a little more justification may help support this statement. For example, the impact of aviation $NO_2$ on $O_3$ and $CH_4$-related species in the upper troposphere and lower stratosphere during the pre-pandemic era have been discussed previously, e.g.*

*Khodayari, A., Tilmes, S., Olsen, S. C., Phoenix, D. B., Wuebbles, D. J., Lamarque, J.-F., and Chen, C.-C.: Aviation 2006 NOx-induced effects on atmospheric ozone and HOx in*

*Community Earth System Model (CESM), Atmos. Chem. Phys., 14, 9925–9939, https://doi.org/10.5194/acp-14-9925-2014, 2014.*

*Khodayari, A, Seth C. Olsen, Donald J. Wuebbles, Daniel B. Phoenix, Aviation NOx-induced CH4 effect: Fixed mixing ratio boundary conditions versus flux boundary conditions, Atmospheric Environment, 113, 135-139, https://doi.org/10.1016/j.atmosenv.2015.04.070, 2015.*

*I think by adding some discussions of these literature may help strengthen your study. In addition, have you tried changing upper tropospheric $CH_4$ in the ECHAM5/MESSy2 model and test its impact on upper tropospheric $O_3$?*

We have added some text on these literature suggestions in the text. Thank you also for the suggestion of investigating the impacts of CH4 on O3 in the UT, which we agree to be an interesting topic to study, but we believe to be beyond the scope of this study.

Lines 342 ff.: The effects of NOx aircraft emissions on O3 and CH4 have been previously discussed for pre-lockdown conditions in Khodayari et al. (2014) and Khodayari et al. (2015) who present increased methane loss rates and a shorter lifetime as a response to increased OH concentrations from aviation as well as higher ozone production rates. Having investigated NOPR in this study, lockdown effects on CH4 loss in the UT induced by reduced air traffic could be subject to future studies.

*Overall, this is a very interesting study. Thank you for your work and good luck!*

*Lin Tan*

*Environmental Sciences, University of California, Riverside*